# Dual-Latent Memory Routing for Vision-Language Reasoning

**Hao-Xuan Ma** [* 1 2]  **Jin-Fei Qi** [* 1 2]  **Yicheng Xiao** [3]  **Han-Jia Ye** [1 2]

## Abstract

Multimodal large language models (MLLMs) have recently made strong progress in vision-language reasoning, yet their performance often degrades as generations grow longer. A key factor is that they frequently lose track of earlier visual evidence and intermediate constraints under a monolithic growing context. Inspired by how humans separately recall *what they see* and *what they infer* when solving complex tasks, we propose **DLMR**, a parameter-efficient mechanism that equips MLLMs with **D**ual **L**atent **M**emories: a *visual memory* that compresses image evidence and a *reasoning memory* that tracks intermediate conclusions and constraints. A **R**outer then dynamically decides which memory and how much to reuse during inference, preserving visual grounding while maintaining coherent long-horizon reasoning. DLMR is trained in three stages, from latent memory construction to selective router learning, while keeping the base MLLM frozen, yielding substantial gains on both general and reasoning benchmarks with only a small number of additional trainable parameters. Analyses further show interpretable, state-dependent routing with specialized memory roles and reduced decoding tokens over long generations. Code is available at https://github.com/Hunter-Wrynn/DLMR.

## 1. Introduction

Multimodal large language models (MLLMs) have recently achieved impressive progress in integrating visual and textual information for complex reasoning tasks (Shao et al., 2024a; Wen et al., 2025). However, as reasoning chains

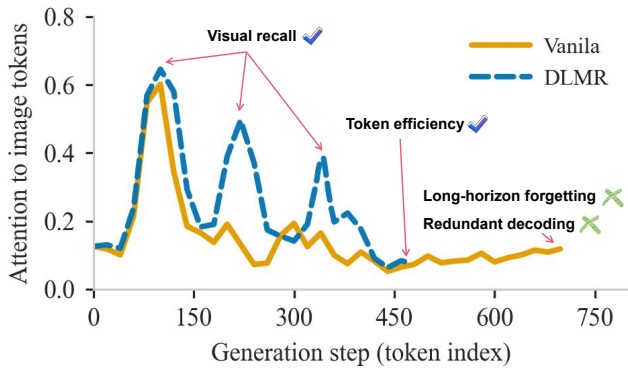

*Figure 1.* Normalized attention to image tokens through generation step. Compared to the vanilla MLLM (Qwen2.5-vl-7B on Math-Vision Benchmark), **DLMR** preserves substantially more visual attention in later decoding, enabling more reliable revisiting of early visual evidence and often requiring fewer reasoning tokens to reach the final answer.

lengthen, MLLMs increasingly fail to preserve early visual evidence and intermediate reasoning states over extended decoding (Sun et al., 2025; Zhou et al., 2025; Tu et al., 2026). A key factor is that most MLLMs rely on a single growing context: the image is encoded once as a prefix, while subsequent states are carried only through generated text (Ren et al., 2025; Xu et al., 2025a). Consistent with this limitation, attention to image tokens steadily decays as generation proceeds (Fig. 1). Prior work encourages revisiting relevant information via prompting-based guidance or training-based alignment (Shao et al., 2024a; Gao et al., 2024; Liu et al., 2025b; Yu et al., 2024b), yet these approaches remain bound to the same growing-context mechanism and thus do not fundamentally prevent long-horizon forgetting—models can still lose effective access to early multimodal cues as decoding length increases.

A natural remedy is to move beyond the single-stream, text-carried context. Cognitive science suggests that robust multi-step reasoning relies on a *disentangled working memory* (Wolpert et al., 2011), where humans keep separate buffers for *observations* and *constraints* and revisit them on demand. Current MLLMs lack a structured alternative to the growing-context paradigm, hindering selective storage and reuse as reasoning progresses. This calls for two architectural ingredients: (i) a *disentangled* representation that can store reusable visual evidence and evolving reasoning constraints, and (ii) a *state-dependent controller* that dy-

---
[*]Equal contribution [1]School of Artificial Intelligence, Nanjing University, China [2]National Key Laboratory for Novel Software Technology, Nanjing University, China [3]Institute of Automation, Chinese Academy of Sciences, Beijing, China. Correspondence to: Han-Jia Ye <yehj@lamda.nju.edu.cn>.

*Proceedings of the 43rd International Conference on Machine Learning*, Seoul, South Korea. PMLR 306, 2026. Copyright 2026 by the author(s).

namically determines *what* and *how much* to retrieve during generation.

Motivated by this, we propose **DLMR**, a parameter-efficient mechanism for long-horizon multimodal reasoning. DLMR instantiates the two ingredients above with **D**ual **L**atent **M**emories and a learned **R**outer: it separates reusable observations from evolving task state, and selectively revisits either source only when needed. This enables *structured, state-dependent* reuse—the model can verify cues from the image when grounding is required, or reuse previously derived state when progressing through multi-step inference, while allocating more computation to harder steps. For instance, in a geometry problem, one step may require checking a diagram attribute, whereas the next step primarily updates previously established relations.

Concretely, DLMR maintains two learnable latent vectors (*visual* and *reasoning*) as disentangled memories. A lightweight *memory injector* converts selected latents into *memory tokens* that are appended during inference, leaving the base MLLM unchanged. A discrete *memory router* then selects which memory to activate and sets a token budget for the current step. As a diagnostic, while the vanilla MLLM rapidly attenuates attention to visual tokens during decoding, DLMR sustains stronger late-stage visual attention with fewer generated tokens on average (Fig. 1).

We also introduce a three-stage training recipe from latent memory construction to selective router learning while keeping the backbone frozen, making DLMR easy to integrate into existing MLLMs with a small parameter overhead. Experiments on challenging long-horizon multimodal reasoning benchmarks show that DLMR improves task performance under controlled inference budgets. Finally, we provide analyses of routing behavior and memory specialization, illustrating when the model prefers visual memory or reasoning memory and how it allocates its thinking budget over the course of generation. In summary, our contributions are as follows:

- We propose *Dual-Latent Memory Routing* (DLMR), a parameter-efficient multimodal memory mechanism that separates reusable perceptual evidence from evolving logical constraints for long-horizon reasoning.

- We design a lightweight memory injector and a memory router that, conditioned on the current context, selects the memory source and token budget for controlled reuse with minimal backbone changes.

- We introduce a three-stage training recipe and provide empirical results and analyses that demonstrate improved long-horizon performance and better ability to revisit visual evidence.

## 2. Related Work

### 2.1. Vision Language Reasoning

Recent multimodal large language models (MLLMs) have substantially improved vision-language reasoning by aligning visual representations with instruction-following language models and scaling up multimodal supervision (Bai et al., 2025; Lu et al., 2025; Agarwal et al., 2025). Building on these foundations, multimodal reasoning is commonly enhanced from three perspectives: **(1) CoT-based** methods elicit multi-step reasoning at inference time via chain-of-thought prompting and test-time scaling, including multimodal variants that ground rationales in visual evidence (Shao et al., 2024a; Xu et al., 2025a; Gao et al., 2024; Jiang et al., 2025; Gao et al., 2025; Wu et al., 2025b); **(2) Training-based** methods improve reasoning through instruction tuning and targeted data curation, further refined by reinforcement learning with task-level feedback or verifiable rewards (Shen et al., 2025; Wen et al., 2025; Wu et al., 2025c; Yu et al., 2024b; Peng et al., 2025); and **(3) RAG-based** methods augment prompts with external evidence (documents, images, or knowledge) to support grounded reasoning and long-context tasks (Kim et al., 2025; Yu et al., 2024a; Ming & Li, 2024; Hao et al., 2025; Tanaka et al., 2025). Despite these advances, long-horizon multimodal reasoning remains challenging as models often struggle to reliably reuse early visual evidence and intermediate constraints over long generations (Sun et al., 2025; Zhou et al., 2025; Tu et al., 2026; Wu et al., 2025a; Li et al., 2025).

### 2.2. Memory Systems

Memory-augmented modeling has been widely studied in LLMs and is increasingly transferred to multimodal settings, where the core goal is to preserve reusable information beyond a single growing context (Hu et al., 2025; Zhang et al., 2025a; Long et al., 2025; Yan et al., 2025). Early designs rely on explicit retrieval, with multimodal variants extending retrieval to images or cross-modal evidence (Kim et al., 2025; Yu et al., 2024a). A complementary direction introduces *latent memory* by learning persistent memory tokens or latent contexts that can be attended to during generation, reducing dependence on long textual traces and enabling parameter-efficient reuse (Yu et al., 2025; Zeng et al., 2025; Zhang et al., 2024b; Liu et al., 2025a). These lines of work motivate memory mechanisms for MLLMs, but existing transfers typically treat memory as a single undifferentiated source—either retrieved context or a monolithic latent space—and provide limited state-dependent control over *what* to reuse and *how much* to think when reasoning unfolds. In contrast, DLMR explicitly separates memory into two specialized latent buffers for visual evidence and reasoning constraints, and employs a discrete router to condition reuse decisions.

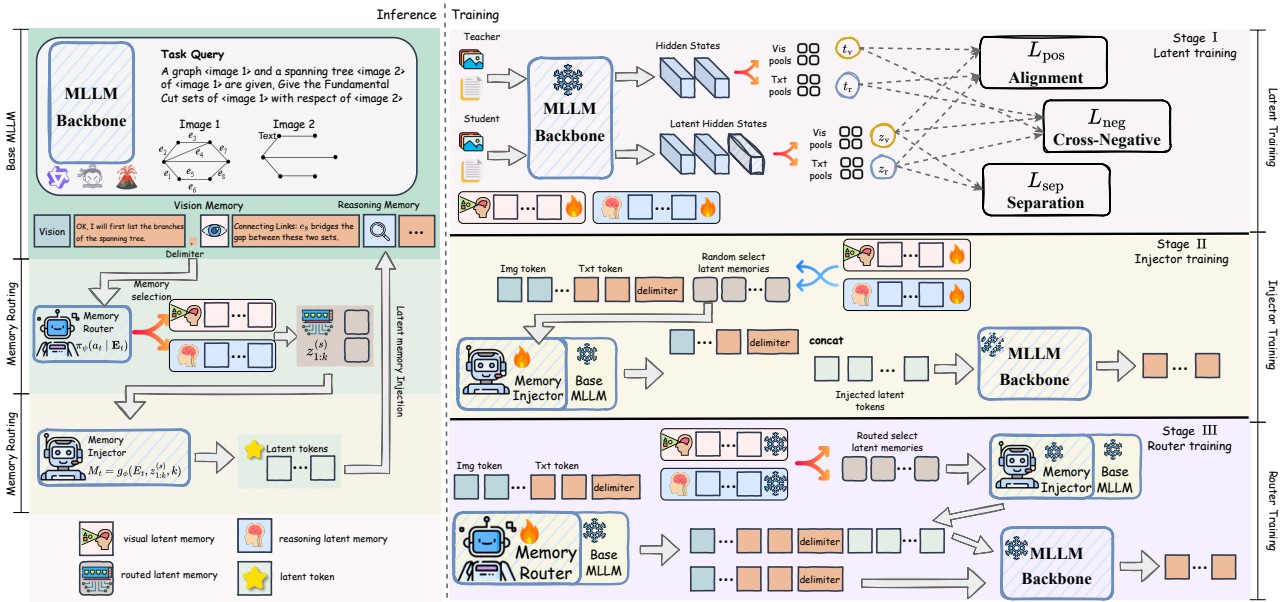

*Figure 2.* **Overview of DLMR.** Given an image $\mathbf{I}$ and text prompt $\mathbf{x}$, a frozen MLLM backbone $\mathcal{M}_\theta$ performs autoregressive decoding. DLMR maintains two compact *latent-space memories* $\mathbf{Z}^{(v)}$ (visual evidence) and $\mathbf{Z}^{(r)}$ (reasoning constraints). At delimiter-eligible steps, a discrete router selects the memory type $s_t \in \{v, r\}$ and an injection budget $k_t \in \mathcal{K}^+$ (or a null action). A lightweight memory injector contextualizes the selected latents into $k_t$ memory tokens $\mathbf{M}_t$, which are appended to the current context before predicting the next token. This enables state-dependent, budget-aware reuse without modifying the backbone.

## 3. Preliminary

**Problem Setup.** We study memory-augmented multimodal reasoning, where each instance consists of an image $\mathbf{I}$ and a tokenized textual instruction (or dialogue history) $\mathbf{x} = (x_1, \ldots, x_L)$. The goal is to generate an answer $\mathbf{y} = (y_1, \ldots, y_T)$ consistent with both modalities. We consider an autoregressive multimodal large language model (MLLM) parameterizing

$$P(\mathbf{y} \mid \mathbf{I}, \mathbf{x}) = \prod_{t=1}^{T} P(y_t \mid \mathbf{I}, \mathbf{x}, \mathbf{y}_{<t}). \quad (1)$$

We build on a Transformer-based MLLM backbone $\mathcal{M}_\theta$ with hidden size $d$ and keep $\theta$ frozen. Following standard MLLMs, $\mathbf{I}$ is encoded into a sequence of visual tokens and concatenated with text tokens to form a unified multimodal input sequence to $\mathcal{M}_\theta$.

**Why long-horizon forgetting arises.** In standard MLLMs, an image $\mathbf{I}$ is encoded into $L_v$ visual tokens $\mathbf{v} = (v_1, \ldots, v_{L_v})$ and concatenated with text prompt $\mathbf{x}$ as a single prefix $\mathbf{s}_0 = [\mathbf{v}; \mathbf{x}]$; the model then generates $\mathbf{y} = (y_1, \ldots, y_T)$ autoregressively. At decoding step $t$, the visible context is $\mathbf{s}_t = [\mathbf{s}_0; \mathbf{y}_{<t}]$ with length $n_t = L_v + L + t - 1$, which grows with $t$. In a decoder-only Transformer, the next-token state is updated by causal self-attention over all previous tokens:

$$\alpha_{t,i} = \frac{\exp(z_{t,i})}{\sum_{j=1}^{n_t} \exp(z_{t,j})}, \qquad z_{t,i} = \langle q_t, k_i \rangle / \sqrt{d}. \quad (2)$$

Since $\sum_{i=1}^{n_t} \alpha_{t,i} = 1$, the *average* attention weight is $1/n_t$; unless the model becomes increasingly peaked on the early visual prefix, the total attention mass on the fixed image-token set $\mathcal{V} = \{1, \ldots, L_v\}$ tends to shrink as the context grows:

$$A_t^{\text{img}} = \sum_{i \in \mathcal{V}} \alpha_{t,i} \approx O\left(\frac{L_v}{n_t}\right) \xrightarrow[t \to \infty]{} 0. \quad (3)$$

Therefore, under a single growing context, early visual evidence becomes progressively harder to reuse, which aligns with long-context retrieval failures (Liu et al., 2024; Hsieh et al., 2024) and the empirically observed *visual grounding decay* in MLLMs, where attention to image tokens weakens as generation proceeds and correlates with hallucinations (Xu et al., 2025b). Motivated by this, DLMR introduces an explicit memory-and-control interface: it stores reusable visual evidence and evolving reasoning constraints in dual latent memories and injects compact memory tokens on demand, mitigating long-horizon forgetting without repeatedly restating state in text.

## 4. Method

As illustrated in Fig. 2, DLMR adds three lightweight components on top of the frozen backbone: (i) dual latent memories for *visual evidence* and *reasoning constraints*, (ii) a memory injector that converts selected latents into step-specific memory tokens, and (iii) a discrete router that de-

cides *when/what/how much* to inject. We describe memory generation in Sec. 4.1, routing in Sec. 4.2, and the training recipe in Sec. 4.3.

## 4.1. Memory Generation

**Dual latent memories.** To enhance long-horizon reasoning without modifying the backbone, we introduce two compact, input-agnostic latent-space memories

$$\mathbf{Z}^{(s)} \in \mathbb{R}^{M_s \times d}, \quad s \in \{v, r\}, \tag{4}$$

where $v$ denotes a *visual* latent-space memory and $r$ denotes a *reasoning* latent-space memory. These memories are shared across all inputs and will be contextualized into step-specific *memory tokens* during decoding. Given a token budget $k \in \mathcal{K}^+$ (e.g., $\{4, 8, 16\}$), we select the first $k$ latent vectors from the chosen memory set, denoted $\mathbf{Z}^{(s)}_{1:k}$.

**Memory injector.** Let $\mathbf{E}_t \in \mathbb{R}^{L_t \times d}$ denote the current multimodal token embeddings (image tokens plus text prefix up to $\mathbf{y}_{<t}$). Conditioned on $(\mathbf{E}_t, \mathbf{Z}^{(s)}_{1:k}, k)$, a lightweight *memory injector* $g_\phi$ contextualizes the selected latents into $k$ memory tokens:

$$\mathbf{M}_t = g_\phi\left(\mathbf{E}_t, \mathbf{Z}^{(s)}_{1:k}, k\right) \in \mathbb{R}^{k \times d}. \tag{5}$$

For the implementation of $g_\phi$, we adopt a parameter-efficient approach by integrating LoRA into a replica of the frozen base MLLM. We first map the current context embeddings to the injector space: $\mathbf{E}^{(w)}_t = P_{\theta \to w}(\mathbf{E}_t)$, and let the latent vectors attend to the context through self-attention. Specifically, we construct the augmented sequence

$$\tilde{\mathbf{E}}^{(w)}_t = [\mathbf{E}^{(w)}_t; \mathbf{Z}^{(s)}_{1:k}] \in \mathbb{R}^{(L_t+k) \times d_w}, \tag{6}$$

and pass it through the injector to obtain hidden states $\tilde{\mathbf{H}}^{(w)}_t$. We then take the last $k$ states as contextualized memory tokens:

$$\mathbf{M}_t = P_{w \to \theta}\left(\tilde{\mathbf{H}}^{(w)}_t[L_t + 1 : L_t + k]\right) \in \mathbb{R}^{k \times d}. \tag{7}$$

These tokens will be appended to the decoding context when the router invokes an injection (Sec. 4.2).

## 4.2. Memory Routing

**Memory-augmented decoding.** We allow memory injection only at a subset of *eligible* generation steps to improve stability and control overhead. Let $\mathcal{T}(\mathbf{y}_{<t}) \subseteq \{1, \ldots, T\}$ denote the eligible set, which may depend on the current decoded prefix. Concretely, we define a small set of delimiter patterns $\mathcal{D}$, and mark step $t$ as *eligible* if the current decoded prefix ends with any delimiter in $\mathcal{D}$. We further cap the number of routed injections by enforcing a per-sample

limit of at most $N_{\max}$ injections. At each $t \in \mathcal{T}(\mathbf{y}_{<t})$, the router chooses a discrete action

$$a_t = (s_t, k_t), \quad s_t \in \{v, r\}, \ k_t \in \mathcal{K}^+, \tag{8}$$

where $\mathcal{K}^+$ is a small set of positive token budgets (e.g., $\{4, 8, 16\}$). If $t$ is not eligible (or the injection cap is reached), we deterministically perform no injection and fall back to standard decoding.

**Routing policy.** We model routing as a lightweight, context-dependent policy $\pi_\psi(a_t \mid \mathbf{E}_t)$. In practice, we implement $\pi_\psi$ as a parameter-efficient LoRA-augmented head on top of the backbone that reuses the hidden states from the ongoing decoding pass. Given the current multimodal prefix, we summarize the decoding state using a compact representation (the final-layer hidden state of the latest prefix token) and map it to a distribution over actions. At inference we use greedy selection for reproducibility, while during router training we sample actions for policy optimization. When the router selects $a_t = (s_t, k_t)$, we generate memory tokens

$$\mathbf{M}_t = g_\phi\left(\mathbf{E}_t, \mathbf{Z}^{(s_t)}_{1:k_t}, k_t\right). \tag{9}$$

We append $\mathbf{M}_t$ to the current context, and then predict the next token with the frozen backbone. This yields a controllable inference-time overhead governed by the injection frequency and budgets $\{k_t\}$.

## 4.3. Training Recipe

We adopt a three-stage training scheme that progressively learns (i) informative latent-space memories, (ii) an effective memory injector, and (iii) an adaptive router. Across all stages, the backbone parameters $\theta$ are kept frozen; we only optimize lightweight memory-related components.

**Stage 1: Latent-space memory pre-warm.** We initialize the latent-space memories $\mathbf{Z}^{(v)}$ and $\mathbf{Z}^{(r)}$ with a self-supervised alignment objective, while keeping the injector and router fixed (or disabled). Given an input $(\mathbf{I}, \mathbf{x})$, we first run the frozen backbone without any routed injection to obtain *teacher* hidden states, and compute pooled representations that summarize (i) the visual evidence from image tokens and (ii) the textual reasoning state from the prefix, denoted $(\mathbf{t}_v, \mathbf{t}_r)$. We then form a *student* pass by injecting latent-space memories at the prompt end (always visual) and at delimiter-eligible points (Sec. 4.2) using randomized memory types $s \in \{v, r\}$ and budgets $k \in \mathcal{K}^+$. From the injected memory tokens we obtain pooled student representations $(\mathbf{z}_v, \mathbf{z}_r)$, and optimize only $\mathbf{Z}^{(v)}$ and $\mathbf{Z}^{(r)}$ via a two-teacher alignment loss:

*Table 1.* Comparison of **DLMR** with strong baselines on **3** general and **4** reasoning benchmarks using two MLLMs (Qwen2.5-VL-7B and InternVL-3-8B). **DLMR (SFT)** trains the injector with supervised learning, while **DLMR (RL)** uses GRPO-based training.

| Method | General | | | | Reasoning | | | | |
|---|---|---|---|---|---|---|---|---|---|
| | MMVet | MMStar | RealWorldQA | Avg | MMMU | MathVerse | MathVision | MathVista | Avg |
| *Qwen2.5-VL-7B* | | | | | | | | | |
| Vanilla | 63.30 | 62.60 | 68.10 | 64.67 | 53.11 | 42.64 | 22.37 | 68.30 | 46.61 |
| CoT | 64.55 | 63.73 | 68.88 | 65.72 | 53.78 | 43.24 | 23.02 | 68.80 | 47.21 |
| CCoT | 64.67 | 64.11 | 69.12 | 65.97 | 53.33 | 43.31 | 24.67 | 67.90 | 47.30 |
| SFT | 64.61 | 62.98 | 69.27 | 65.62 | 54.16 | 43.38 | 23.68 | 68.90 | 47.53 |
| GRPO | 68.06 | 65.33 | 70.41 | 67.93 | 57.74 | 45.39 | 28.33 | 69.70 | 50.29 |
| Visual-RFT | 68.77 | 65.27 | 71.09 | 68.38 | 58.54 | 45.27 | 29.12 | 70.10 | 50.76 |
| RCTS-RAG | 66.02 | 65.56 | 69.74 | 67.11 | 55.27 | 44.76 | 26.94 | 69.80 | 49.19 |
| **DLMR** SFT | **74.77** | 68.26 | **71.33** | **71.45** | 61.00 | 46.95 | 35.32 | 72.10 | 53.84 |
| **DLMR** RL | 73.85 | **68.67** | 70.70 | 71.07 | **62.77** | **48.98** | **39.34** | **74.70** | **56.45** |
| *InternVL-3-8B* | | | | | | | | | |
| Vanilla | 75.08 | 68.46 | 70.98 | 71.51 | 57.22 | 37.44 | 27.30 | 71.10 | 48.27 |
| CoT | 76.98 | 70.13 | 72.04 | 73.05 | 58.44 | 42.13 | 33.55 | 71.90 | 51.51 |
| CCoT | 78.02 | 71.87 | 73.16 | 74.35 | 58.28 | 41.06 | 32.14 | 71.80 | 50.82 |
| SFT | 77.63 | 69.74 | 72.75 | 73.37 | 58.28 | 41.06 | 32.14 | 71.80 | 50.82 |
| GRPO | 81.55 | 72.85 | 73.21 | 75.87 | 62.11 | 45.71 | 35.81 | 73.70 | 54.33 |
| Visual-RFT | 82.17 | 72.51 | 73.43 | 76.04 | 63.78 | 46.25 | 35.43 | 74.60 | 55.02 |
| RCTS-RAG | 78.29 | 71.62 | 72.37 | 74.09 | 58.94 | 41.57 | 33.97 | 72.10 | 51.65 |
| **DLMR** SFT | **86.32** | 76.54 | **74.88** | **79.25** | **67.45** | 49.31 | 37.18 | 77.20 | 59.04 |
| **DLMR** RL | 85.91 | **77.10** | 74.02 | 79.01 | 66.33 | **52.14** | **38.56** | **81.30** | **63.08** |

$$\mathcal{L}_{\text{stage1}} = \sum_{s \in \{v,r\}} \Big( 1 - \cos(\mathbf{z}_s, \mathbf{t}_s) \Big)$$
$$+ \lambda_{\text{neg}} \sum_{s \neq s'} \Big[ \cos(\mathbf{z}_s, \mathbf{t}_{s'}) - m \Big]_+ \quad (10)$$
$$+ \lambda_{\text{sep}} \big| \cos(\mathbf{z}_v, \mathbf{z}_r) \big|,$$

where $\lambda_{\text{neg}}$ weights a cross-teacher hinge term (margin $m$) that penalizes mismatched alignment, and $\lambda_{\text{sep}}$ weights a separation term that discourages similarity between $\mathbf{z}_v$ and $\mathbf{z}_r$.

**Stage 2: Injector training.** We train the injector parameters $\phi$ together with the latent-space memories $\mathbf{Z} = \{\mathbf{Z}^{(v)}, \mathbf{Z}^{(r)}\}$ on downstream multimodal reasoning data, while keeping the router disabled. To make both memories robust under different overhead constraints, we expose the model to mixed injections (e.g., alternating or random) by varying the memory type $s \in \{v, r\}$ and budget $k \in \mathcal{K}^+$ at eligible points during training. We optimize either (i) supervised next-token likelihood when gold outputs are available, or (ii) GRPO when only task-level feedback is available. The objective can be written compactly as

$$\max_{\phi, \mathbf{Z}} \mathbb{E}_{(\mathbf{I}, \mathbf{x}, \mathbf{y}) \sim \mathcal{D}} \big[ \log P_{\theta, \phi, \mathbf{z}}(\mathbf{y} \mid \mathbf{I}, \mathbf{x}) \big] - \epsilon \mathcal{L}_{\text{preserve}}, \quad (11)$$

where $\theta$ is frozen and $\mathcal{L}_{\text{preserve}}$ is a weak specialization regularizer to prevent the two memories from drifting toward a single shared representation (we use a small cross-branch

hinge term plus a separation term, consistent with Stage 1).

**Stage 3: Router learning.** Finally, we freeze the injector and latent-space memories, and train the router policy $\pi_\psi$ to adaptively choose $(s_t, k_t)$ at eligible steps (Sec. 4.2) using GRPO. To balance accuracy and efficiency, we optimize a cost-aware objective:

$$\max_{\psi} \mathbb{E}_{\tau \sim \pi_\psi} \big[ R^{\text{task}}(\tau) + \lambda_{\text{eff}} R^{\text{eff}}(\tau) \big] - \beta \, \text{KL}(\pi_\psi \| \pi_{\text{ref}}),$$
$$(12)$$

where $\tau$ denotes a sampled decoding trajectory (including routing actions), and $\pi_{\text{ref}}$ is a fixed reference router for stabilization. In our implementation, $R^{\text{eff}}(\tau)$ encourages smaller average injection budgets and is only counted when the answer is correct (see Appendix B).

## 5. Experiments

### 5.1. Experimental Setup

**Benchmarks.** We evaluate **DLMR** on a diverse set of vision-language benchmarks that cover both general multimodal understanding and long-horizon reasoning. For general capability, we report results on **MMVet** (Yu et al., 2023), **MMStar** (Chen et al., 2024), and **RealWorldQA**. For reasoning-centric evaluation, we use **MMMU** (Yue et al., 2024), **MathVerse** (Zhang et al., 2024a), **MathVision** (Wang et al., 2024a), and **MathVista** (Lu et al., 2023). Details are in Appendix C.3.

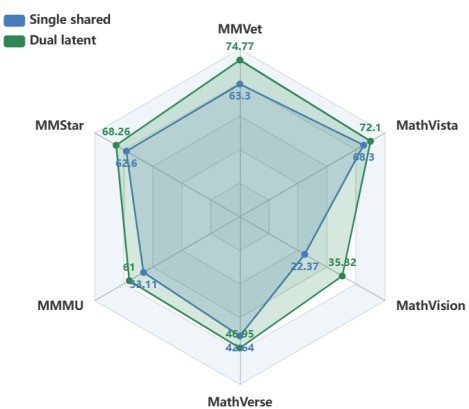

*Figure 3.* Disentanglement ablation on **Qwen2.5-VL-7B** evaluated on six benchmarks (two general: MMVet, MMStar; four reasoning: MMMU, MathVerse, MathVision, MathVista). We compare using the dual latent memory ($\mathbf{Z}^{(v)}, \mathbf{Z}^{(r)}$) and a single shared memory ($\mathbf{Z}$).

**Baselines.** We compare **DLMR** against a set of strong baselines spanning three common paradigms: **(i) CoT-based** inference-time prompting, including standard **CoT** and **CCoT** (Cheng & Van Durme, 2024); **(ii) training-based** post-training methods, including **SFT**, **GRPO** (Shao et al., 2024b), and **Visual-RFT** (Liu et al., 2025b); and **(iii) RAG-based** augmentation represented by **RCTS-RAG** (Yang et al., 2025). Details are in Appendix C.2.

**Implementation Details.** All experiments are implemented on Qwen2.5-VL-7B (Bai et al., 2025) and InternVL3-8B (Zhu et al., 2025). During the three-stage training procedure, we use the same training dataset. Initially, we include the training split dataset of the selected benchmarks and retain their original data division. For benchmarks without a training phase, we use them solely for evaluation. Additionally, we incorporate the OpenMMReasoner dataset (Zhang et al., 2025b), improving the reasoning. Notably, since Stage 2 admits two training variants (SFT and GRPO), we also report corresponding baseline variants trained with SFT or RL.

### 5.2. Main Results

We first present the main results showing that **DLMR** improves both general VQA and reasoning benchmarks (Tab. 1). We then conduct three ablations to identify the source of gains: **[Abl.1]** removes memory disentanglement, which collapses $\{\mathbf{Z}^{(v)}, \mathbf{Z}^{(r)}\}$ into a single latent set $\mathbf{Z}$ (Fig. 3); **[Abl.2]** disables *injector training* by keeping the injector $g_\phi$ non-trainable while retaining the same injection interface (Tab. 2); and **[Abl.3]** disables adaptive routing via fixed budgets $k \in \{4, 8, 16\}$ to examine the accuracy–cost trade-off (Tab. 4).

*Table 2.* Injector ablation on **Qwen2.5-VL-7B** evaluated on four reasoning benchmarks. We compare the full model with a *frozen injector* variant (no injector training) while keeping the same dual memories and routing/injection interface.

| Variant | MMMU | MVerse | MathV | MVista | Avg |
|---|---|---|---|---|---|
| Frozen injector | 58.14 | 44.32 | 27.74 | 69.96 | 50.44 |
| Trainable injector | 61.00 | 46.95 | 35.32 | 72.10 | 53.84 |

**DLMR enables reliable long-horizon multimodal reasoning.** As shown in Tab. 1, **DLMR** consistently outperforms prompt-based reasoning, post-training, and retrieval-based baselines, and the gains cannot be attributed solely to stronger prompting or additional parameter updates. Instead, DLMR introduces an explicit memory interface that *separates* reusable visual evidence from evolving intermediate constraints and *selectively reuses* them during generation. The improvements are most pronounced on long-horizon reasoning benchmarks (e.g., MathVision/MathVista), where conventional methods often lose early visual cues or drift from intermediate constraints as decoding proceeds; by revisiting the appropriate memory stream on demand, DLMR mitigates such drift and yields larger gains in precisely these challenging regimes. Meanwhile, DLMR also improves general VQA performance, suggesting no trade-off between general perception and reasoning. Empirically, **DLMR (SFT)** performs best on general benchmarks, while **DLMR (RL)** achieves the strongest reasoning results (Tab. 1), indicating that SFT stabilizes broad visual grounding whereas RL further strengthens multi-step decision-making and constraint maintenance.

**[Abl.1] Disentanglement is crucial.** We ablate memory disentanglement on Qwen2.5-VL-7B by collapsing the dual buffers into a single shared latent memory $\mathbf{Z}$, and comparing it with the full dual-latent design ($\mathbf{Z}^{(v)}, \mathbf{Z}^{(r)}$) on six benchmarks (two general and four reasoning). As shown in Fig. 3, the dual-latent design consistently outperforms the shared-memory variant, improving the overall six-benchmark average from **52.05** to **59.73** (**+7.68**); restricted to the four reasoning benchmarks, the gain widens from **46.61** to **53.84** (**+7.23**). Notably, the largest single-benchmark gain occurs on the most long-horizon benchmark *MathVision* (22.37→**35.32**), where models are more prone to losing early visual evidence or drifting from intermediate constraints, suggesting that separating *visual evidence* and *intermediate constraints* directly addresses the key failure mode we target. In contrast, collapsing both information types into a single undifferentiated buffer limits specialization and can introduce interference, leading to weaker and less stable performance across tasks.

**[Abl.2] A learnable injector is necessary.** We freeze the *memory injector* while keeping the same dual memories and injection interface (Tab. 2). This variant still appends latent

*Table 3.* **Cross-backbone generalization of DLMR** on three additional MLLMs (LLaVA-OV-1.5-4B, Qwen2.5-VL-3B, InternVL-3-2B), evaluated on General and Reasoning benchmarks. ↑ indicates the performance improvement over the corresponding base model. DLMR yields consistent gains across all three backbones, indicating that the proposed memory augmentation is largely base-model agnostic.

| Base Model | General | | | | Reasoning | | | | |
|---|---|---|---|---|---|---|---|---|---|
| | MMVet | MMStar | RealWorldQA | Avg | MMMU | MathVerse | MathVision | MathVista | Avg |
| LLaVA-OV-1.5-4B | 66.0 | 62.6 | 68.1 | 64.6 | 56.0 | 42.6 | 22.3 | 68.3 | 46.6 |
| + DLMR | 75.1 ↑9.1 | 68.9 ↑6.3 | 71.3 ↑3.2 | 71.4 ↑6.8 | 63.9 ↑7.9 | 46.9 ↑4.3 | 35.3 ↑13.0 | 72.1 ↑3.8 | 53.8 ↑7.2 |
| Qwen2.5-VL-3B | 61.2 | 55.1 | 65.3 | 60.6 | 51.2 | 31.2 | 21.9 | 61.2 | 41.4 |
| + DLMR | 69.8 ↑8.6 | 60.6 ↑5.5 | 70.1 ↑4.8 | 66.8 ↑6.2 | 58.9 ↑7.7 | 33.2 ↑2.0 | 36.5 ↑4.6 | 68.6 ↑7.4 | 49.3 ↑7.9 |
| InternVL-3-2B | 65.1 | 61.3 | 64.5 | 63.6 | 48.6 | 24.1 | 21.7 | 54.7 | 37.3 |
| + DLMR | 70.4 ↑5.3 | 66.1 ↑4.8 | 69.5 ↑4.0 | 68.7 ↑5.1 | 55.3 ↑6.7 | 30.3 ↑6.2 | 29.1 ↑8.4 | 62.1 ↑7.4 | 44.2 ↑6.9 |

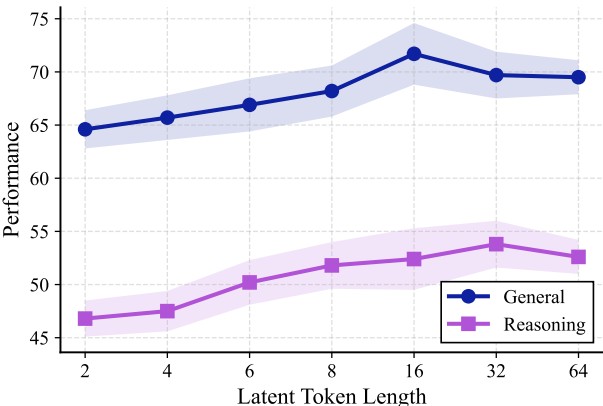

*Figure 4.* Accuracy–length frontier under different router budget caps $k_{\max}$ on Qwen2.5-VL-7B. **General** tasks peak at $k_{\max} = 16$, while **Reasoning** tasks favor a larger headroom and peak at $k_{\max} = 32$, indicating that adaptive routing uses high budgets only on hard states.

*Table 4.* Budget ablation (**Qwen2.5-VL-7B**). We compare fixed injection budgets $k$ (no adaptive routing) with the full DLMR router. **Avg Acc** is averaged over four reasoning benchmarks; **Avg token length** is the average number of generated tokens per sample.

| Variant | Avg Acc | Avg token length |
|---|---|---|
| No router, fixed $k=4$ | 51.55 | **664** |
| No router, fixed $k=8$ | 52.71 | 732 |
| No router, fixed $k=16$ | 52.04 | 765 |
| Full DLMR (adaptive $k_t$) | **53.84** | 677 |

memories and retrieves the corresponding hidden states as memory tokens, but it can no longer adapt how these latents are contextualized into task-useful signals. Freezing the injector causes a clear performance drop on reasoning benchmarks relative to the full model, showing that *where* memory is injected is insufficient—a *trainable interface* is needed to translate latent buffers into actionable, context-aligned tokens. We attribute the degradation to a contextualization mismatch: without training, injected tokens become less informative (or partially redundant), weakening the model's ability to reliably revisit visual evidence and maintain intermediate constraints in long generations. Overall, this ablation confirms that DLMR's gains rely on both disentangled memories and a learnable injector that grounds and shapes memory tokens for downstream reasoning.

**[Abl.3] Adaptive routing yields a better accuracy–cost frontier.** We replace state-dependent budget allocation with a fixed injection budget $k \in \{4, 8, 16\}$ at every eligible insertion point, while keeping the same dual memories and injector. As shown in Tab. 4, larger fixed bud-

gets substantially increase generation length (avg tokens: $664 \rightarrow 732 \rightarrow 765$ for $k = 4, 8, 16$), yet the accuracy is not monotonic. In particular, $k = 8$ achieves the best fixed-budget accuracy (52.71), whereas $k = 16$ produces longer outputs but slightly worse accuracy (52.04), indicating diminishing returns and possible redundancy from over-injection. We further vary the router's maximum allowable budget $k_{\max}$ and report the resulting accuracy and cost in Fig. 4. Interestingly, the optimal frontier differs by task family: **General** benchmarks achieve their best trade-off at $k_{\max} = 16$, whereas **Reasoning** benchmarks benefit from a larger budget headroom and peak at $k_{\max} = 32$. This suggests that general VQA queries rarely require aggressive memory injection, while multi-step reasoning problems more often profit from occasionally allocating a larger budget on hard states, with the router still selecting small budgets on easy ones.

### 5.3. Detailed Analysis.

We provide additional analyses to better understand **when** and **why** DLMR helps. Specifically, we analyze (i) whether DLMR brings *backbone-agnostic* gains on general vision–language benchmarks, (ii) whether the learned memory routing mechanism *transfers* to *text-only* reasoning, and (iii) whether DLMR improves *token efficiency* by reducing redundant long-form generations. More analyses on router interpretability, wall-clock inference overhead, and transfer to unseen benchmarks are provided in the appendix A.

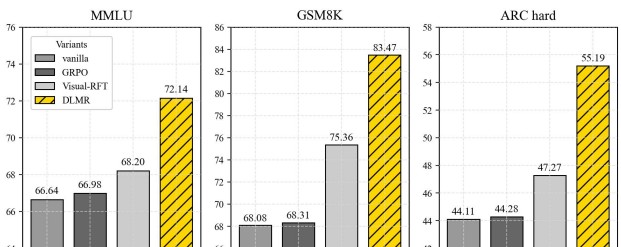

*Figure 5.* Cross-modality transfer of memory routing to text-only benchmarks (MMLU, GSM8K, ARC-Challenge) on **Qwen2.5-VL-7B**, compared against the vanilla backbone, GRPO, and Visual-RFT baselines.

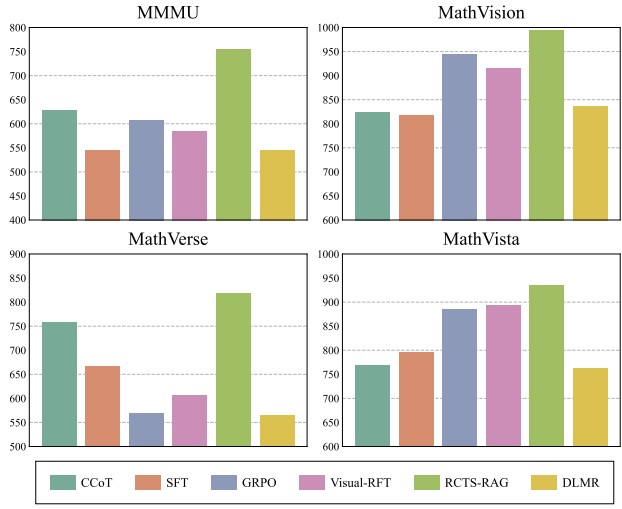

*Figure 6.* Average generated tokens per sample on four reasoning-centric benchmarks using **Qwen2.5-VL-7B**. By selectively reusing latent memories, DLMR shortens generations compared to strong baselines.

**[Ana.1] Model generalization.** Tab. 3 shows that DLMR yields consistent gains across three distinct base MLLMs (LLaVA-OV-1.5-4B, Qwen2.5-VL-3B, and InternVL-3-2B), indicating that the proposed memory augmentation is largely base-model agnostic. On *general* VQA benchmarks, DLMR improves the average score by +6.8 on LLaVA-OV-1.5-4B, +6.2 on Qwen2.5-VL-3B, and +5.1 on InternVL-3-2B, with consistent gains on MMVet, MMStar, and RealWorldQA. More importantly, DLMR delivers even larger improvements on *reasoning*-centric benchmarks, boosting the average by +7.2, +7.9, and +6.9, respectively; the gains are particularly pronounced on MMMU and MathVision, suggesting that explicitly modeling reusable visual evidence and evolving reasoning constraints benefits long-horizon multimodal reasoning across diverse base MLLMs.

**[Ana.2] Cross-modality transfer of memory routing.** Although DLMR is motivated by vision-language reasoning, its core idea—separating reusable context from evolving constraints and enabling state-dependent, budget-aware reuse—is modality-agnostic. To test transfer beyond multimodal inputs, we apply the same routing and injection mechanism to text-only benchmarks (MMLU (Hendrycks et al., 2020), GSM8K (Cobbe et al., 2021) and ARC-Challenge (Clark et al., 2018)) without any vision features. Benchmark details are in Appendix C.3. As shown in Fig. 5, DLMR consistently improves over the vanilla backbone and a GRPO baseline, with the largest gains on multi-step reasoning tasks. In particular, DLMR achieves +15.39 on GSM8K (68.08→83.47) and +11.08 on ARC-Challenge (44.11→55.19), indicating that the learned latent memories and router can preserve and reuse intermediate reasoning state even in purely textual settings. Overall, these results suggest that DLMR provides a transferable mechanism for more reliable long-horizon decoding.

**[Ana.3] Improved token efficiency via selective memory reuse.** We measure token efficiency as the average number of generated tokens per sample on reasoning benchmarks. For completeness, we include the tokens used to generate the latent memories (both visual and reasoning) when computing DLMR's total generated length. As shown in Fig. 6, DLMR produces shorter or comparable generations to strong baselines and achieves the best overall average token efficiency. In particular, DLMR reduces average generated tokens by 9.7% compared to Visual-RFT (the closest competitor in capability), and by 4.2% compared to SFT (the closest baseline in token usage), indicating a more compact decoding trajectory by replacing redundant textual re-derivation with selective latent revisitation.

## 6. Conclusion

We presented DLMR, a parameter-efficient memory augmentation mechanism for long-horizon vision–language reasoning. DLMR equips a frozen MLLM with dual latent memories that disentangle reusable visual evidence from evolving reasoning constraints, and couples them with a lightweight injector plus a discrete, state-dependent router that selects *what* to reuse and *how much* to inject at each step. With a simple three-stage training recipe, DLMR delivers consistent gains on both general VQA and reasoning benchmarks across multiple backbones, with the largest improvements on tasks that demand reliable visual revisiting and constraint maintenance over long generations. Beyond multimodal settings, DLMR transfers to text-only reasoning and improves token efficiency, while the learned router exhibits interpretable, budget-aware behavior that concentrates computation on harder states.

## Acknowledgements

This work was supported in part by the National Key Research and Development Program of China (No. 2024YFE0202800), the Basic Research Program of Jiangsu under Grant No. BK20253021, the National Natural Science Foundation of China (NSFC) under Grants No. 62522605 and No. 62376118, the Fundamental and Interdisciplinary Disciplines Breakthrough Plan of the Ministry of Education of China (No. JYB2025XDXM118), the "111 Center" (No. B26023), and the Collaborative Innovation Center of Novel Software Technology and Industrialization.

## Impact Statement

This paper presents Dual-Latent Memory Routing (DLMR), a parameter-efficient memory augmentation for multimodal large language models (MLLMs) that aims to improve long-horizon vision–language reasoning by storing reusable visual evidence and intermediate reasoning constraints in disentangled latent memories and selectively re-injecting them during generation. If deployed responsibly, DLMR can benefit applications such as education and tutoring on visual problems, accessibility tools that require grounded image understanding, and scientific/engineering assistants that must preserve intermediate constraints across long solutions; selective reuse may also reduce unnecessary generation, lowering latency and energy under fixed inference budgets. However, improved grounding and long-horizon reasoning can also increase capability in harmful settings (e.g., surveillance enablement or more convincing misinformation), and stronger-looking outputs may be over-trusted despite remaining hallucinations, biases, or brittleness under distribution shift inherited from the backbone and data. As with many vision–language methods, data provenance may raise privacy or copyright concerns depending on training and deployment practices. We position DLMR as a research mechanism rather than an autonomous decision system for high-stakes use, and recommend standard safeguards in any deployment, including human oversight, dataset documentation and bias auditing, privacy-preserving data handling, and safety filtering/policy enforcement to mitigate misuse.

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

This supplementary material provides additional details supporting the main paper, organized as follows:

- **Section A**: **Additional Experiments and Analyses.** Extended experimental results and ablations beyond those reported in the main paper.

- **Section B**: **Method details.** Additional formulations and algorithmic descriptions of the proposed routing framework, including objective definitions and design rationales that complement the main text.

- **Section C**: **Training details.** Full training configurations and hyperparameters, data preprocessing, optimization settings, and implementation choices to ensure reproducibility.

## A. More experiments

**Interpretable routing behavior.** Fig. A1 visualizes the router's *injection frequency* over the relative generation position, separately aggregated on reasoning benchmarks (top) and general benchmarks (bottom). A clear and intuitive pattern emerges: on *reasoning* tasks, the router invokes the **reasoning memory** more frequently than the visual memory, whereas on *general* tasks it preferentially activates the **visual memory**. This aligns with the differing information demands across task families: multi-step reasoning requires repeatedly revisiting intermediate constraints and partial conclusions, while general perception-heavy queries benefit more from re-accessing detailed visual evidence. Beyond the overall preference, the curves also exhibit *position-dependent* dynamics: both memories are injected more often in the earlier and middle stages of decoding and gradually taper off near the end, suggesting that the router concentrates memory reuse when establishing key evidence/constraints and relies less on injection once the answer becomes determinate.

**Inference efficiency on the benchmark suite.** Tab. A1 reports end-to-end inference time and accuracy aggregated over the General and Reasoning benchmark suites, evaluated on two backbones. Accuracy is the average over the three General benchmarks (MMVet, MMStar, RealWorldQA) and the four Reasoning benchmarks (MMMU, MathVerse, MathVision, MathVista); time is the average end-to-end inference latency per sample. Across both models, **DLMR** consistently improves accuracy while maintaining (or reducing) inference time relative to strong SFT baselines. On *Qwen2.5-VL-7B*, DLMR improves general accuracy from 65.6% to 71.4% while slightly reducing time (5.6→5.4s), and yields a larger gain on reasoning (47.5%→53.8%) together with a notable speedup (14.0→11.5s, ≈18%). On *InternVL3-8B*, DLMR similarly boosts accuracy on both general (73.4%→79.3%) and reasoning (50.8%→59.0%)

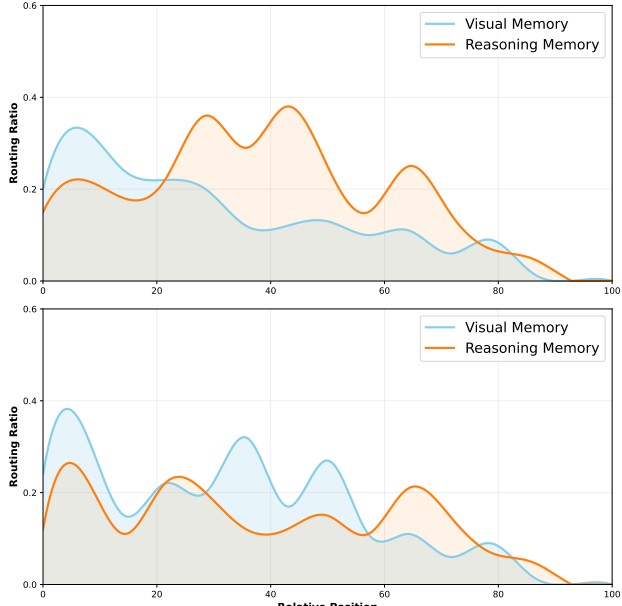

*Figure A1.* **Router frequency over generation position.** We plot the routing ratio (injection frequency) of the visual memory and the reasoning memory as a function of relative decoding position, aggregated over reasoning benchmarks (top) and general benchmarks (bottom). The router invokes **reasoning memory** more often on reasoning tasks, while it prefers **visual memory** on general tasks, indicating task-adaptive and interpretable routing.

with comparable latency to SFT (3.5 vs. 3.7s; 13.4 vs. 13.1s). These results suggest that DLMR's selective latent reuse can *reduce redundant token-level re-derivation* in long generations: it preserves or improves wall-clock efficiency while strengthening correctness, especially on multi-step reasoning where repeated constraint maintenance would otherwise inflate decoding time.

**Transfer to unseen benchmarks.** Tab. A2 evaluates whether the learned memory routing generalizes beyond the training distribution on *unseen* benchmarks, covering a reasoning dataset (LOGICVISTA) and two perception/OCR datasets (OCR-BENCH, MMT-BENCH). We compare against prompting baselines (Vanilla, CoT) and training-based baselines (SFT, GRPO). Overall, **DLMR** achieves the best performance on all three unseen datasets, improving LOGICVISTA to 55.0 (vs. 50.2 for GRPO), OCR-BENCH to 92.0 (vs. 90.1 for SFT), and MMT-BENCH to 62.5 (vs. 58.0 for GRPO), indicating that the disentangled memories transfer to both reasoning and perception/OCR tasks outside the training distribution.

## B. Methodology Details

This appendix provides additional implementation and modeling details for DLMR to complement Secs. 4.1–4.3.

*Table A1.* **Inference time and accuracy on the General/Reasoning benchmark suites.** Accuracy is averaged over the three General benchmarks (MMVet, MMStar, RealWorldQA) and the four Reasoning benchmarks (MMMU, MathVerse, MathVision, MathVista) from Tab. 1; time is the average end-to-end inference time per sample.

| Model & Method | General | | Reasoning | |
|---|---|---|---|---|
| | Time (s) | Acc (%) | Time (s) | Acc (%) |
| *Qwen2.5-VL-7B* | | | | |
| Vanilla | 10.1 | 64.7 | 28.5 | 46.6 |
| SFT | 5.6 | 65.6 | 14.0 | 47.5 |
| **DLMR** | 5.4 | 71.4 | 11.5 | 53.8 |
| *InternVL3-8B* | | | | |
| Vanilla | 8.4 | 71.5 | 21.1 | 48.3 |
| SFT | 3.5 | 73.4 | 13.4 | 50.8 |
| **DLMR** | 3.7 | 79.3 | 13.1 | 59.0 |

*Table A2.* **Transfer to unseen benchmarks.** We evaluate models trained under the default setting on **unseen** datasets spanning reasoning (LOGICVISTA) and general perception/OCR (OCR-BENCH, MMT-BENCH). All numbers are task scores in % (higher is better).

| Method | LOGICVISTA | OCR-BENCH | MMT-BENCH |
|---|---|---|---|
| Vanilla | 43.1 | 86.1 | 51.5 |
| CoT | 47.5 | 89.1 | 55.0 |
| SFT | 49.0 | 90.1 | 56.5 |
| GRPO | 50.2 | 88.4 | 58.0 |
| **DLMR** | 55.0 | 92.0 | 62.5 |

## B.1. Injection Eligibility and Budget Cap

**Delimiter-based eligibility.** We restrict routed injections to a subset of steps to improve stability and control overhead. Let $\mathcal{D}$ be a small delimiter set (e.g., $\mathcal{D} = \{\texttt{,}, \texttt{.}, \texttt{\textbackslash n}\}$). At decoding step $t$, we mark the step as *eligible* if the decoded prefix $\mathbf{y}_{<t}$ ends with any pattern in $\mathcal{D}$, i.e.,

$$t \in \mathcal{T}(\mathbf{y}_{<t}) \iff \exists d \in \mathcal{D} : \mathbf{y}_{<t} \text{ ends with } d. \quad (13)$$

If $t \notin \mathcal{T}(\mathbf{y}_{<t})$, we deterministically perform no injection.

**Per-sample injection cap.** To avoid excessive overhead, we cap the number of routed injections to at most $N_{\max}$ per sample. Once the cap is reached, the router is no longer queried and decoding proceeds without further injections.

## B.2. Injection Placement

**Fixed visual injection at prompt end.** In addition to routed injections during decoding, we apply a fixed prompt-end injection using visual latents ($s = v$) to provide stable perceptual cues. Importantly, for chat-style MLLMs, we insert the prompt-end memory tokens *before* the assistant marker (rather than after it). This keeps the final token immediately preceding generation a real text token from the

prompt, while still exposing the injected memory tokens to the subsequent autoregressive pass.

## B.3. Router Action Space and Parameterization

**Discrete action encoding.** Let $\mathcal{K}^+ = \{k^{(1)}, \ldots, k^{(m)}\}$ be the candidate budget set. We define a routed action set $\mathcal{A} = \{v, r\} \times \mathcal{K}^+$ with $|\mathcal{A}| = 2m$ actions. At each eligible step, the router outputs either a null decision (no injection) or an action $a_t = (s_t, k_t) \in \mathcal{A}$. For implementation convenience, we index actions by a single integer $a \in \{0, \ldots, 2m - 1\}$:

$$s(a) = \begin{cases} v, & a < m, \\ r, & a \geq m, \end{cases} \qquad k(a) = k^{(1 + (a \bmod m))}. \quad (14)$$

**Policy network.** We parameterize the routing policy as $\pi_\psi(a_t \mid \mathbf{E}_t)$ using a lightweight classifier head on top of the backbone hidden states. Concretely, at step $t$ we take the final-layer hidden state of the most recent prefix token (or an equivalent pooled representation) and map it to $2m$ logits via a linear layer. We optionally enable LoRA adapters on the router branch for parameter efficiency while keeping the backbone weights $\theta$ frozen.

**Sampling vs. greedy.** During router training, we sample actions from $\pi_\psi$ to enable policy optimization. At inference time, we use greedy action selection for reproducibility.

## B.4. Memory Injector Architecture

**Injector as a LoRA-augmented copy.** We implement the injector $g_\phi$ as a copy of the backbone with LoRA adapters enabled in the injector branch. Given current context embeddings $\mathbf{E}_t \in \mathbb{R}^{L_t \times d}$, selected latents $\mathbf{Z}_{1:k}^{(s)} \in \mathbb{R}^{k \times d}$, and budget $k$, we form

$$\tilde{\mathbf{E}}_t^{(w)} = [P_{\theta \to w}(\mathbf{E}_t); \mathbf{Z}_{1:k}^{(s)}] \in \mathbb{R}^{(L_t + k) \times d_w}, \quad (15)$$

run the injector Transformer to obtain $\tilde{\mathbf{H}}_t^{(w)}$, and extract the last $k$ states as contextualized memory tokens. If $d_w \neq d$, we use learned linear projections $P_{\theta \to w}$ and $P_{w \to \theta}$ to map between embedding spaces.

**Cache invalidation.** When memory tokens are injected mid-generation, the effective prefix changes. In practice we reset the key–value cache after an injection to ensure correctness, and continue decoding from the augmented prefix.

## B.5. Training Details

**Stage 1: teacher–student pooling.** We compute teacher representations $(\mathbf{t}_v, \mathbf{t}_r)$ by running the frozen backbone without injection and pooling hidden states over (i) image-token positions and (ii) non-image (text) positions, respectively. We compute student representations $(\mathbf{z}_v, \mathbf{z}_r)$ by pooling the injected memory-token hidden states corresponding

to the visual and reasoning latent banks. We use a masked mean pooling:

$$\text{pool}(\mathbf{H}, \mathbf{m}) \triangleq \frac{\sum_i m_i \mathbf{H}_i}{\sum_i m_i}, \tag{16}$$

where $\mathbf{m}$ is a binary mask over sequence positions.

**Stage 1: injection mixing and budget coverage.** To make both banks usable under different compute budgets, in the student pass we always inject visual latents at the prompt end, and at delimiter-eligible points we alternate (or randomly sample) the latent type $s \in \{v, r\}$ and the budget $k \in \mathcal{K}^+$. This matches the inference-time interface while avoiding dependence on any supervised output labels.

**Stage 2: preserve regularizer.** During injector training, we optionally add a weak specialization regularizer

$$\begin{aligned}
\mathcal{L}_{\text{preserve}} \triangleq &\left[ \cos(\mathbf{z}_v, \mathbf{t}_r) - m \right]_+ \\
&+ \left[ \cos(\mathbf{z}_r, \mathbf{t}_v) - m \right]_+ \\
&+ \left| \cos(\mathbf{z}_v, \mathbf{z}_r) \right|,
\end{aligned} \tag{17}$$

which mirrors Stage 1 by discouraging cross-branch confusion and collapse while allowing task learning to dominate.

**Stage 3: GRPO for routing.** We freeze the injector and both latent banks, and optimize the router with GRPO. For each input, we sample a group of trajectories under the current policy, compute a task reward and an efficiency reward, and form group-relative advantages for policy updates. We include a KL penalty to a reference policy $\pi_{\text{ref}}$ for stabilization, and only query the router at delimiter-eligible steps until the injection cap is reached.

**Efficiency reward.** We define $R^{\text{eff}}$ to prefer smaller budgets and apply it only when the model is correct: for example, if $k_{\max} = \max(\mathcal{K}^+)$ and $\bar{k}$ is the average injected budget over the trajectory, we use

$$R^{\text{eff}}(\tau) = \mathbb{I}\{R^{\text{task}}(\tau) = 1\} \cdot \left(1 - \bar{k}(\tau)/k_{\max}\right). \tag{18}$$

## C. Training Details

This section describes full training configurations, data preprocessing, optimization, and implementation choices used in our experiments. We omit the specific backbone identity (model name / parameter count) since our method is backbone-agnostic.

### C.1. Configuration and Reproducibility

**Configuration system.** All experiments are specified by a YAML configuration and optional command-line overrides. Key model-level knobs include the prompt-end injection

length $M_v$, the per-step routed injection length $M_r$, the budget set $\mathcal{K}^+$, and the maximum number of routed injections $N_{\max}$.

**Seeding.** We fix a global random seed (default: 42) for Python, NumPy, and PyTorch, and enable deterministic CuDNN where applicable. For stochastic training procedures (e.g., GRPO rollouts), we keep sampling enabled and report average results across the evaluation split.

**Precision and distributed training.** We train with mixed-precision activations (BF16 where supported) and use the `accelerate` launcher with DeepSpeed ZeRO stage-2 for data parallel training. We keep the backbone frozen throughout; only memory-related parameters are trainable.

**Backbone architectures.** Tab. A3 summarizes the key architectural hyperparameters of the two backbone MLLMs used in our main experiments.

### C.2. Baselines

We select a total of 7 baselines, including the **vanilla** model; 2 CoT-based inference-time prompting methods: **CoT**, **CCoT** (Cheng & Van Durme, 2024); 3 direct training-based post-training methods: **SFT**, **GRPO** (Shao et al., 2024b), **Visual-RFT** (Liu et al., 2025b); 1 RAG-based augmentation method: **RCTS-RAG** (Yang et al., 2025).

All the baselines are implemented on two base models: Qwen2.5-VL-7B (Bai et al., 2025) and InternVL3-8B (Zhu et al., 2025). For strategies initially implemented on other base models, e.g. GPT-4o (Hurst et al., 2024) and Qwen2-VL(Wang et al., 2024b), we transfer them to Qwen2.5-VL-7B (Bai et al., 2025) and InternVL3-8B (Zhu et al., 2025) for fair comparison.

### C.3. Data Format and Preprocessing

**Benchmarks.** We evaluate on a diverse suite of vision-language benchmarks spanning general multimodal understanding and long-horizon reasoning, plus three text-only benchmarks for cross-modality transfer. Tab. A4 summarizes the benchmarks; we follow each benchmark's official evaluation protocol.

**Dataset splits.** For datasets that provide official train/validation/test splits, we keep the original splits unchanged. For benchmarks that do not provide a training split (or are intended as evaluation-only), we use them solely for evaluation. For datasets used for training but lacking a predefined validation set, we reserve a small portion of the training split for validation. We optionally subsample a fixed number of examples for quick ablations.

**Training corpus.** Across the three-stage training procedure, we use a unified training corpus that (i) aggregates the available training splits of the selected benchmarks and (ii) in-

*Table A3.* **Backbone model configurations.** Key architectural hyperparameters of the two backbone MLLMs used in our experiments/analysis: Qwen2.5-VL-7B-Instruct and InternVL3-8B.

| Configurations | Parameters | Qwen2.5-VL-7B | InternVL3-8B |
|---|---|---|---|
| Text backbone | hidden_size $d$ | 3584 | 3584 |
| | num_layers $L$ | 28 | 28 |
| | num_heads $H$ | 28 | 28 |
| | num_kv_heads $H_{\mathrm{kv}}$ | 4 | 4 |
| | mlp_intermediate $d_{\mathrm{ff}}$ | 18944 | 18944 |
| | max_position_emb | 128000 | 32768 |
| | rope_theta | $10^6$ | $10^6$ |
| Vision backbone | vision_hidden_size | 1280 | 1024 |
| | vision_num_layers | 32 | 24 |
| | vision_num_heads | 16 | 16 |
| | vision_mlp_intermediate | 3420 | 4096 |
| | image_size | 448 | 448 |
| | patch_size | 14 | 14 |
| Tokenization | vocab_size | 152064 | 151674 |
| | dtype / attn_impl | bfloat16 / SDPA | bfloat16 / SDPA |

*Table A4.* **Evaluation benchmarks.** #Eval refers to the size of the split we evaluate on.

| Benchmark | Domain | # Eval | Answer fmt. | Metric |
|---|---|---|---|---|
| *General (vision–language)* | | | | |
| MMVet (Yu et al., 2023) | Composite VL skills | 218 | Open-ended | LLM-judge acc. |
| MMStar (Chen et al., 2024) | Vision-indispensable VL | 1,500 | MCQ | Accuracy |
| RealWorldQA | Real-world VQA | 765 | MCQ / short ans. | Accuracy |
| *Reasoning (vision–language)* | | | | |
| MMMU (Yue et al., 2024) | College-level multidisciplinary | 900 (val) | MCQ / open | Accuracy |
| MathVerse (Zhang et al., 2024a) | Visual math (diagram variants) | 3,940 (testmini) | Open-ended | Accuracy |
| MathVision (Wang et al., 2024a) | Competition math w/ figures | 3,040 | Open-ended | Accuracy |
| MathVista (Lu et al., 2023) | Visual math (testmini) | 1,000 | MCQ / open | Accuracy |
| *Text-only (cross-modality transfer)* | | | | |
| MMLU (Hendrycks et al., 2020) | Knowledge & reasoning (57 subjects) | 14,042 | MCQ | Accuracy |
| GSM8K (Cobbe et al., 2021) | Grade-school math word problems | 1,319 | Open-ended | Exact match |
| ARC-Challenge (Clark et al., 2018) | Hard science QA | 1,172 | MCQ | Accuracy |

corporates the OpenMMReasoner dataset to strengthen reasoning skills. Stage 2 admits two variants (SFT and GRPO), and we report corresponding baseline variants trained with SFT or RL on the same training corpus and with matched decoding configurations when applicable.

**Multimodal input representation.** Each example provides either (i) a plain-text instruction prompt or (ii) a chat-style message list with interleaved text and image references. Images are loaded from a dataset-specific root directory and passed to the MLLM processor to obtain visual tensors. If an example references a missing image and a blank placeholder image is available in the dataset directory, we fall back to the placeholder for robust training; otherwise we raise an error.

**Tokenization and supervision masks (SFT).** We build a full training sequence using the backbone chat template, and construct labels by masking out: (i) all tokens that belong to the prompt (only the completion is supervised), (ii) padding tokens, and (iii) special vision placeholder tokens used by

the MLLM tokenizer. This yields standard next-token cross-entropy on the assistant completion tokens.

**Length filtering.** To avoid out-of-memory issues and unstable delimiter injection, we filter out samples whose tokenized prompt length exceeds a stage-specific maximum length. For stage-1 cold-start we set a larger maximum length (e.g., 2048 tokens); for GRPO we constrain the prompt length by `max_prompt_length`.

## C.4. Model and Injection Hyperparameters

**Latent banks and budgets.** We maintain two learnable latent banks with sizes $M_v$ (visual) and $M_r$ (reasoning). At each injection, we select the first $k$ latents from the chosen bank with $k \in \mathcal{K}^+$. We use a small discrete budget set, typically $\mathcal{K}^+ = \{4, 8, 16\}$.

**Injection cap and eligibility.** We cap the number of routed injections per sample to $N_{\max}$ (e.g., $N_{\max} = 5$). Router queries are restricted to delimiter-eligible steps defined by a small delimiter set $\mathcal{D}$ (e.g., punctuation and newline).

**LoRA configuration (injector and router).** We use parameter-efficient LoRA adapters for both the injector branch and the router branch. Unless stated otherwise, we use rank $r = 16$, scaling $\alpha = 32$, dropout $0.1$, and apply LoRA to attention projection modules (e.g., query/key/value projections) while keeping all backbone weights frozen.

## C.5. Stage 1: Latent Cold-Start

**Trainable parameters.** We optimize only the two latent banks $\mathbf{Z}^{(v)}$ and $\mathbf{Z}^{(r)}$; LoRA adapters and router parameters

are disabled.

**Objective and pooling.** We compute teacher representations from a forward pass without injection and pool hidden states over image-token positions and non-image positions to obtain $(\mathbf{t}_v, \mathbf{t}_r)$. We compute student representations $(\mathbf{z}_v, \mathbf{z}_r)$ by pooling over injected memory-token positions corresponding to each bank. We optimize the two-teacher alignment loss in Eq. (6) with hyperparameters $(\lambda_{\text{neg}}, \lambda_{\text{sep}}, m)$.

**Injection schedule during cold-start.** We always inject visual latents at the prompt end. At delimiter-eligible points, we alternate (or randomly sample) the memory type $s \in \{v, r\}$ and budget $k \in \mathcal{K}^+$ to ensure both banks learn to function under different compute budgets.

**Optimization.** We use AdamW with a cosine learning-rate schedule and linear warmup. A typical setting is: learning rate $10^{-3}$, warmup ratio $0.1$, batch size 1 (per-device), and 3 epochs. We disable gradient checkpointing in this stage for stability with multimodal processors, but it can be enabled if memory becomes a bottleneck.

## C.6. Stage 2: Injector Training (SFT/GRPO)

**Trainable parameters.** We enable the injector LoRA adapters and jointly optimize injector parameters $\phi$ and latent banks $\mathbf{Z}$. The router is disabled.

**SFT settings.** We optimize next-token cross-entropy on completion tokens with AdamW and cosine schedule. Typical settings include: learning rate $10^{-5}$, warmup ratio $0.1$, BF16, maximum sequence length 1024–2048, and stage-specific batch sizes with gradient accumulation to match the available hardware.

**Mixed injections during training.** To match inference-time behavior, we inject at the prompt end and at delimiter-eligible points. At eligible points, we mix visual and reasoning injections (alternating or random) and sample budgets $k \in \mathcal{K}^+$.

**Specialization preserve regularizer.** Optionally, we apply a small-weight regularizer to discourage the two banks from collapsing into the same representation. In our implementation, this corresponds to a hinge-based cross-branch penalty with margin $m$ plus an inter-bank separation term, scaled by $\epsilon$ (e.g., $\epsilon = 10^{-3}$).

**GRPO settings (if used for injector learning).** When training the injector with GRPO, we generate multiple rollouts per prompt (`num_generations`), compute group-relative advantages, and include an optional KL penalty to a reference policy for stabilization. We set generation sampling enabled during training (nonzero temperature) to ensure reward variance within each prompt group.

## C.7. Stage 3: Router Training (GRPO)

**Trainable parameters.** We freeze the injector and latent banks and optimize only the router parameters $\psi$ (classification head and optional router LoRA adapters).

**Action space.** The router predicts a discrete action over $\mathcal{A} = \{v, r\} \times \mathcal{K}^+$ at eligible steps; non-eligible steps use a null action.

**GRPO hyperparameters.** We use group sampling with `num_generations` rollouts per prompt and optimize a cost-aware objective that adds an efficiency reward favoring smaller budgets when the answer is correct. We cap prompt length (`max_prompt_length`) and completion length (`max_completion_length`) for stable training and bounded compute. We use AdamW with cosine schedule, BF16, and gradient accumulation as needed.

**Implementation notes.** Since routed injections modify the effective prefix, we reset the decoding key–value cache after each injection and continue generation from the augmented sequence. We also recommend inserting prompt-end memory tokens before the assistant marker in chat-formatted prompts to avoid degenerate early termination.

*Table A5.* **Hyperparameters across three training stages.** Stage 1: latent cold-start (optimize latent banks only). Stage 2: injector SFT (PEFT, backbone frozen). Stage 3: router RL (GRPO, backbone frozen). Values shared across both backbones (Qwen2.5-VL-7B and InternVL3-8B) unless stated otherwise. Batch sizes are per-device; gradient accumulation is used when needed.

| Configurations | Parameters | Stage 1 | Stage 2 | Stage 3 |
|---|---|---|---|---|
| Core | backbone | Qwen2.5-VL-7B / InternVL3-8B | Qwen2.5-VL-7B / InternVL3-8B | Qwen2.5-VL-7B / InternVL3-8B |
| | visual_latents_len $M_v$ | 8 | 8 | – |
| | reasoning_latents_len $M_r$ | 16 | 16 | – |
| | max_visual_aug_num | 1 | 1 | 1 |
| | max_reasoning_aug_num $N_{\max}$ | 5 | 5 | 5 |
| | random_reasoning_aug | True | – | – |
| | reasoning_latents_source | – | alternate | – |
| | reasoning_latents_budgets | – | [4, 8, 16] | – |
| | reasoning_latents_visual_prob | – | 0.5 | – |
| | action_budget_set | – | – | [4, 8, 16] |
| | num_actions $|\mathcal{A}|$ | – | – | 6 (Visual/Reasoning $\times$ {4,8,16}) |
| | router_trainset | – | – | Random 50% subset of train split |
| | router_validset | – | – | Random 50% subset of valid split |
| Loss | neg_weight | 1.0 | – | – |
| | sep_weight | 0.1 | – | – |
| | margin | 0.2 | – | – |
| Augmentation | delimiter_set | [, , ., \n] | – | – |
| | budget_set | [4, 8, 16] | – | – |
| | branch_choice | alternate (visual/reasoning) | – | – |
| Preserve | sft_preserve_epsilon | – | 0.001 | – |
| | sft_preserve_margin | – | 0.2 | – |
| LoRA | rank $r$ | 16 | 16 | 16 |
| | $\alpha$ | 32 | 32 | 32 |
| | drop_out_rate | 0.1 | 0.1 | 0.1 |
| | target_module | [q_proj, v_proj] | [q_proj, v_proj] | [q_proj, v_proj] |
| Training | method | Self-supervised alignment | SFT | GRPO |
| | max_length | 2048 | 2048 | – |
| | max_prompt_length | – | – | 1024 |
| | max_completion_length | – | – | 1024 |
| | batch_size | 1 | – | – |
| | per_device_batch_size | – | 1 | 4 |
| | grad_accum | 1 | 1 | – |
| | epoch | 3 | 8 | 2 |
| | num_generations (group size) | – | – | 8 |
| | num_iterations | – | – | 1 |
| | loss_type | – | – | BNPO |
| | kl_coefficient $\beta$ | – | – | 0.0 |
| | efficiency_reward_weight $\lambda_{\text{eff}}$ | – | – | 1.0 |
| | temperature | – | – | 1.0 |
| | learning_rate | $1 \times 10^{-3}$ | $1 \times 10^{-5}$ | $1 \times 10^{-5}$ |
| | warmup_ratio | 0.1 | 0.1 | 0.1 |
| | optimizer | AdamW | AdamW | AdamW |
| | scheduler | Cosine | Cosine | Cosine |
| | precision | bf16 | bf16 | bf16 |
| | deepspeed | ZeRO-2 | ZeRO-2 | ZeRO-2 |
| | attention_impl | SDPA | SDPA | SDPA |

