# OpenReview forum: "Dual-Latent Memory Routing for Vision-Language Reasoning"
_ICML.cc/2026/Conference — ICML 2026 spotlight_

### Official Review · Reviewer_7Qnx · 2026-02-14

**Soundness:** 3
**Presentation:** 3
**Significance:** 3
**Originality:** 2
**Overall Recommendation:** 4
**Confidence:** 5

**Summary:**

The paper proposes "Dual-Latent Memory Routing" (DLMR) to address the "visual forgetting" and reasoning degradation observed in MLLMs during long-horizon tasks. The core hypothesis is that monolithic context windows dilute attention to early visual cues. The authors propose a parameter-efficient solution involving (1) two global, learnable latent banks (visual vs. reasoning), (2) a "memory injector" (a LoRA-adapted copy of the backbone) that contextualizes these latents, and (3) a discrete router that dynamically inserts these memory tokens at delimiter-specific steps. The method is trained via a three-stage process involving alignment, injector training, and GRPO-based router optimization.

**Compliance With Llm Reviewing Policy:**

Affirmed.

**Key Questions For Authors:**

- Global vs. Local: Since $Z$ is shared across all inputs, how do you ensure the "Visual Memory" is not just learning dataset priors (e.g., "most charts have an x-axis") rather than helping recall the specific image provided?
- Injector Cost: Does the Injector ($g_\phi$) require a full forward pass of the backbone (with LoRA) for the current context length $L_t$? If so, is the FLOPs cost effectively double that of the vanilla model for every step where injection occurs?
- Text-Only Transfer: If "Visual Memory" is designed to compress image evidence, why does the method yield such strong gains on text-only GSM8K? Does this imply the "Visual" bank is actually just a second "Reasoning" bank?

**Limitations:**

- The approach depends on delimiter patterns and injection caps, potentially harming robustness under different prompting styles or open-ended dialogue.
- Dual memories are shared across all inputs, which may limit instance-specific fidelity and raise concerns about spurious memorization or dataset-specific shortcuts.
- Efficiency claims based on token counts may not translate to end-to-end latency given router + injector overhead.
- The evaluation story would benefit from stronger controls on training compute parity, data overlap/leakage, and robustness to format shifts.

**Strengths And Weaknesses:**

Strengths:
- Architectural Clarity: The separation of "static/perceptual" evidence and "evolving/reasoning" constraints is a logical architectural prior. The interface between the router and injector is well-defined.
- Empirical Performance: The method demonstrates consistent gains across a wide variety of benchmarks, particularly on complex reasoning suites like MathVision and MathVista, and shows strong cross-model generalization (Qwen, InternVL).
- Token Efficiency: The analysis showing that DLMR reduces the average number of generated tokens (Fig. 6) is a tangible benefit for long-horizon generation.

Weaknesses:
- Conceptual Disconnect in "Memory" Definition (Major)The paper frames the method as a "Memory" system, but the implementation relies on input-agnostic, global latent vectors ($Z^{(v)}, Z^{(r)}$) shared across all samples [Section 4.1]. True "memory" in the context of reasoning usually implies the retention of instance-specific history (like a KV cache or RAG retrieval). Here, $Z$ functions more like global soft-prompts or "learnable query tokens" (similar to DETR queries or Perceiver latents).While the Injector conditions these on the current context ($E_t$), the "memory banks" themselves do not store information from the image or the specific problem instance. This raises concerns that the model is simply learning dataset-specific biases or "reasoning macros" in these global vectors, rather than a mechanism to actually "recall" specific visual pixels from the start of the prompt.
- "Visual Forgetting" vs. General Reasoning CapacityThe motivation heavily relies on "visual forgetting" due to attention dilution. However, the successful transfer to text-only benchmarks (GSM8K, ARC in Fig. 5) undermines this narrative. If the method provides substantial gains on text-only math problems where there is no "early visual evidence" to forget, it suggests the mechanism functions as a generic "compute extender" or "latent scratchpad" rather than a solution specific to multimodal grounding. The "Visual Memory" bank may simply be acting as additional capacity for general processing, making the "Dual-Latent" distinction less theoretically distinct than claimed.
- Architectural Overhead and Efficiency TransparencyWhile Appendix A (Table A1) reports favorable wall-clock times due to reduced generation length, the architectural cost per step is understated in the main text. The Memory Injector is implemented as a replica of the base MLLM with LoRA [Section 4.1]. If I understand correctly, every time an injection is triggered, the model must perform a forward pass through this injector (processing context $E_t$). This implies a significant FLOPs overhead per injected step compared to a standard forward pass. The trade-off between "fewer tokens generated" and "heavier compute per injected step" requires more transparent analysis (e.g., FLOPs per sample) rather than just latency, which can be influenced by system optimizations.
- Brittleness of Delimiter-Based RoutingThe routing mechanism is gated by hard-coded delimiter patterns (e.g., newlines, punctuation) [Section 4.2]. This design introduces a fragility: if the base model's prompting style changes or if it fails to emit standard delimiters (common in diverse chat templates), the memory mechanism is never triggered. This dependency on output formatting limits the robustness of the method compared to a fully learnable, dense gating mechanism.
- Training Complexity vs. GainsThe proposed recipe is complex, involving three distinct training stages and RL (GRPO). It is difficult to disentangle how much of the performance gain comes from the specific "Dual-Latent" architecture versus the benefits of simply applying GRPO/RL and extensive post-training to the base model. The baselines include GRPO, but the specific tuning of the DLMR pipeline (Stage 1 alignment + Stage 2 mixing + Stage 3 RL) might offer optimization advantages that obscure the contribution of the architecture itself.

---

> ### Author Rebuttal · Authors · 2026-03-31
>
> We thank the reviewer for the thoughtful and constructive feedback.
>
> ### Response to Q1 / W1
> We agree that DLMR is not an external or episodic memory in the retrieval sense: the latent banks $Z^{(v)}$ and $Z^{(r)}$ are global, input-agnostic slots shared across samples. More precisely, DLMR provides an **instance-conditioned latent memory interface**, where selected latents are transformed into **instance- and step-specific memory tokens** by the injector $g_\phi(E_t, Z^{(s)}_{1:k}, k)$ conditioned on the current multimodal prefix.
>
> This is also why DLMR is not equivalent to static soft prompts. If the banks only captured dataset priors, a frozen injector should remain largely sufficient; however, freezing the injector causes a clear drop (Abl2; Table 2), showing that the gain comes from context-conditioned transformation rather than static latent prompts alone. And collapsing the dual banks into a single shared latent set substantially hurts performance (Abl1; Fig. 3), indicating that the benefit does not come from generic extra capacity, but from the structured separation between visual evidence and reasoning constraints.
>
> |Condition|Full DLMR|Visual bank disabled|
> |---|---:|---:|
> |Original Image|72.1|69.0|
> |Randomly Swapped|38.6|37.8|
> |Blank Image|24.9|24.5|
>
> To directly address the dataset-prior concern, we add a test on MathVista with Qwen2.5-VL: we keep the question fixed and replace the image with either a randomly swapped image or a blank image (Table R1). The key signal is not just that performance drops under image corruption, but that the **advantage of Full DLMR over the visual-bank-disabled variant shrinks sharply**. If the visual bank mainly encoded dataset-level priors, this gap would remain largely intact. Its collapse under image corruption instead supports that the visual branch depends on the specific input image through context-conditioned reuse.
>
> ### Response to Q2 / W3
> We agree that the per-injection cost should be stated more explicitly. When routing is triggered, the injector performs an additional forward over the current prefix plus the selected latents. However, this does not mean the overall decoding cost is doubled at every step: routed injections are **sparse rather than dense**, since they are only allowed at delimiter-eligible steps and are further capped by $N_{\max}$, while all non-triggered steps follow standard decoding. Thus, the total overhead is better characterized as the sum of injector forward + cache rebuild over a **small number of routed events**.
>
> Importantly, cache-rebuild cost scales with the current prefix length, which is typically much larger in later decoding stages; this explains why cache rebuild can exceed injector-forward time in long-horizon settings. To make this trade-off explicit, we report a decomposed efficiency analysis in **Table R2**.
>
> |Benchmark|length|injections|budget|inj(s)|cache(s)|decode(s)|total(s)|
> |---|---:|---:|---:|---:|---:|---:|---:|
> |MathVista|702|2.8|7.2|0.76|1.35|9.34|11.45|
>
> ### Response to Q3 / W2
> **Visual forgetting is the motivating multimodal failure mode, while the underlying mechanism is a more general interface for long-horizon state reuse**. In multimodal reasoning, this state often includes early visual evidence; in text-only reasoning, it mainly consists of intermediate constraints and partial conclusions. Thus, the text-only gains do not contradict our motivation, they show that the routed latent-reuse mechanism generalizes beyond the visual case.
>
> At the same time, these results do not imply that the visual bank is simply a redundant second reasoning bank. If the two branches were functionally identical, collapsing them into a single shared latent pool should be nearly harmless; empirically, it causes a substantial drop, indicating that structured dual-bank separation matters. Moreover, the learned router already shows asymmetric usage of the visual or reasoning branches across task families, which is inconsistent with full functional redundancy.
>
> | Benchmark | Modality | Visual ratio \(p_v\) |
> |---|---|---:|
> |GSM8K|Text-only|0.04|
> |MMStar|Multimodal|0.46|
>
> We further analyze the **visual-memory routing ratio** on Qwen2.5-VL. As shown in **Table R3**, the routing statistics exhibit clear **modality-dependent specialization**: the router almost never activates the visual branch in the absence of visual input, but uses it much more frequently when visual grounding is required. Therefore, the strong text-only gains are primarily attributable to the **reasoning branch** and the broader long-horizon state-reuse mechanism, rather than to the visual bank acting as a redundant second reasoning bank.

---

> > ### Author Rebuttal · Reviewer_7Qnx · 2026-04-01
> >
> > Thank you for the detailed and thoughtful rebuttal. I appreciate the authors’ efforts in addressing several of my concerns.
> >
> > In particular, the additional analyses regarding the role of the dual latent banks are helpful. The injector ablation, dual-bank collapse, and image corruption experiments provide more convincing evidence that the proposed mechanism is not merely capturing dataset-level priors, but instead leverages instance-conditioned information. The routing statistics across modalities are also useful in clarifying that the visual and reasoning branches are not trivially redundant. These additions significantly strengthen the empirical support of the method.
> >
> > The clarification on efficiency, including the breakdown of injector and cache-related costs, is also appreciated and improves transparency compared to the original submission.
> >
> > However, some concerns remain. The conceptual framing of the method as a “memory” mechanism is still somewhat ambiguous, given that the latent banks are shared across inputs and do not explicitly store instance-specific history in the conventional sense. While the rebuttal clarifies this design choice, the distinction from global latent prompts or learned scratchpads could be better articulated in the paper.
> >
> > In addition, the reliance on delimiter-based routing raises potential robustness issues under different prompting formats or generation styles, which remains insufficiently addressed. The complexity of the training pipeline (multi-stage training with RL) also makes it difficult to fully disentangle the contribution of the architectural design from optimization effects.
> >
> > Overall, the rebuttal meaningfully improves the clarity and empirical support of the work, although some conceptual and robustness-related concerns remain.

---

> > > ### Author Response · Authors · 2026-04-02
> > >
> > > Thank you for the thoughtful follow-up. We are encouraged that the added evidence on the dual latent banks, instance-conditioned reuse, cross-modality routing behavior, and efficiency breakdown has strengthened the empirical support and improved the transparency of the paper. We also appreciate your constructive remaining concerns on the conceptual framing, routing robustness, and training complexity, and we address them below.
> > >
> > > ### Response to the remaining concern on "memory"
> > > Thank you for this thoughtful clarification. A more precise characterization is that DLMR provides an **instance-conditioned latent working-memory interface** for selective reuse during decoding.
> > > Concretely, the latent banks are global and input-agnostic at the parameter level, but they are **not used as static latent prompts**. Thus, while the latent slots are shared, the effective memory tokens used by the model are context-specific. This is also the key distinction from global latent prompts or learned scratchpads.
> > >
> > > In particular, we will clarify that "memory" in DLMR should be understood as a **working-memory-style latent interface** for reactivating reusable information during generation, rather than as a claim of explicit sample-specific storage.
> > >
> > >
> > > ### Response to the delimiter-based routing concern
> > > Thank you for raising this important point. We note that a related concern was also raised by Reviewer tVLa (Weakness 1). We agree that delimiter-based routing should be viewed as a **practical eligibility policy** for deciding when the router is queried, rather than a core assumption of the method. The main architectural contribution remains the combination of **dual latent banks, a contextual injector, and state-dependent routing over memory type and budget**; delimiter-triggering is simply a lightweight mechanism for reducing routing frequency and controlling overhead.
> > >
> > > Our motivation for using delimiters is that they often correspond to relatively stable local boundaries in generation (e.g., punctuation or line breaks), making them a simple and low-cost point at which to reconsider whether additional memory reuse is needed. Importantly, we do not claim that this is the only valid triggering strategy.
> > >
> > > To assess robustness, we added a sensitivity analysis (also included in our response to Reviewer tVLa) comparing **no gating**, the **default delimiter set** \(D=\{, . \n\}\), and a **coarser rule** \(D=\{\n\}\) as Table R1. These results suggest that the method is **not brittle to a single prompting format or generation style**; rather, delimiter-based routing mainly provides a favorable balance between routing frequency, generation length, and final accuracy. We will clarify this in the revision and explicitly note that other eligibility policies.
> > >
> > > | Delimiter | MathVista | Avg.Len | MMStar | Avg.Len |
> > > |---|---:|---:|---:|---:|
> > > | No gating | 69.9 | 942 | 64.7 | 661 |
> > > | Coarser \(D=\{\n\}\) | 70.2 | 722 | 67.1 | 524 |
> > > | **Default \(D=\{, . \n\}\)** | **72.1** | 746 | **68.2** | 533 |
> > >
> > >
> > >
> > >
> > > ### Response to the concern about training complexity and optimization effects
> > >
> > > Thank you for this important point. We note that a related concern was also raised by Reviewer KSkE (Weakness 1 and Question 1).
> > > We agree that the current training pipeline is not minimal, and that it is valuable to distinguish architectural contributions from optimization effects more clearly. Our claim is therefore not that the multi-stage recipe is uniquely necessary, but rather that it is a stable way to train the proposed architecture while keeping the backbone frozen.
> > >
> > >
> > > At the same time, the current evidence suggests that the gains are **not primarily an optimization artifact**.
> > > In out ablation, we make test on (1) collapsing the two memories into one shared bank, (2) freezing the injector, (3) replacing adaptive routing with fixed budgets and for rebuttal to Reviewer KSkE, we make ablation on (4) remove Stage 1 and randomly initialize the latent banks. All these results indicate that importance of the architectural design itself, rather than optimization alone.
> > >
> > > We will revise the paper to make this separation clearer. In particular, we will present the three stages as an effective training strategy for learning different parts of the architecture—latent specialization, contextualized injection, and adaptive routing.

---

### Official Review · Reviewer_KSkE · 2026-03-01

**Soundness:** 2
**Presentation:** 2
**Significance:** 2
**Originality:** 2
**Overall Recommendation:** 4
**Confidence:** 3

**Summary:**

This paper strives to outline a central context regarding the limitation of current Multimodal Large Language Models (MLLMs) in long-horizon reasoning: the tendency to lose track of early visual evidence and intermediate constraints due to a monolithic growing context. To address this, the authors propose **DLMR (Dual-Latent Memory Routing)**, a parameter-efficient mechanism that augments a frozen MLLM backbone. DLMR introduces two disentangled latent memory banks—one for visual evidence and one for reasoning constraints. A lightweight "Memory Injector" contextualizes these latents, and a learnable "Router" dynamically decides when and which memory to inject into the context stream during decoding. The system is trained via a three-stage pipeline involving latent alignment, injector training, and router reinforcement learning (GRPO). Overall, the paper's major theme concerns establishing a structured, state-dependent memory reuse mechanism that improves performance on complex reasoning benchmarks while enhancing token efficiency.

**Compliance With Llm Reviewing Policy:**

Affirmed.

**Final Justification:**

I think rebuttal solved my concerns. I change my score from 3 to 4.

**Key Questions For Authors:**

1.   Is the full three-stage training strictly necessary? Specifically, how much performance drop is observed if Stage 1 (Latent Pre-warm) is skipped and the latent banks are initialized randomly and trained jointly with the injector in Stage 2?
2.  As noted in the Appendix, injecting memory tokens requires cache invalidation/resetting. Could you provide a more detailed comparison of the *end-to-end wall-clock latency* (not just token counts) compared to the baseline, particularly for the longer-context benchmarks? Does the overhead of the Router + Injector + Cache Reset outweigh the speedup from reduced token generation?
3.  The paper states the latent memories $Z^{(s)}$ are "input-agnostic" (Eq. 4) and shared across all inputs. This implies they act as global "registers" rather than instance-specific memory buffers. How does the model prevent these global latents from becoming saturated or biased towards the training distribution, thereby hurting generalization on unseen OOD tasks?
4.   Have you compared the "Reasoning Memory" injection against a simpler baseline where the router simply injects learnable, static "pause tokens" (without the injector architecture looking back at the context)? This would clarify if the gain is from the *memory mechanism* or simply from allocating more compute steps.

**Limitations:**

yes

**Strengths And Weaknesses:**

**Strengths:**

*   The core idea of disentangling "what is seen" (visual memory) from "what is inferred" (reasoning memory) is cognitively plausible and architecturally sound. Moving away from a purely linear context growth to a selective injection mechanism effectively addresses the "long-horizon forgetting" problem.
*   A significant strength is the observation that DLMR achieves higher accuracy with fewer generated tokens (Fig. 6). By injecting compressed memory tokens, the model avoids redundant textual re-derivation of established facts.

**Weaknesses:**

*   The proposed three-stage training recipe (Latent Pre-warm $\rightarrow$ Injector Training $\rightarrow$ Router Learning) is quite complex. While each stage has a justification, the cumulative engineering burden is high. It is unclear from the main text how sensitive the final performance is to the hyperparameters of the early stages (e.g., the alignment loss in Stage 1).
*  While *token efficiency* is highlighted, the paper mentions in the Appendix that the KV cache is reset after memory injection to ensure correctness. This operation can be computationally expensive in terms of wall-clock time, potentially offsetting the gains from generating fewer tokens. The main paper discusses "efficiency" mostly in terms of token count, which might be slightly misleading regarding real-world latency.
*  The "Reasoning Memory" effectively acts as learnable "thinking tokens." It is difficult to disentangle whether the gain comes specifically from the *content* of the reasoning memory or simply from the additional computation depth provided by injecting extra tokens (similar to "pause tokens" or "thinking tokens" in recent LLM literature).

---

> ### Author Rebuttal · Authors · 2026-03-31
>
> We thank the reviewer for the constructive comments on training complexity, efficiency, and the interpretation of the reasoning memory.
>
> ### Response to W1/Q1
> **Regarding the necessity of the three-stage recipe,** we do not claim it is the only viable training path. Rather, **Stage 1 provides useful preconditioning** for the dual latent banks before injector training: it initializes $Z^{(v)}$ and $Z^{(r)}$ with alignment, cross-negative, and separation losses so that they specialize toward visual evidence and reasoning state. Since Stage 2 uses only a weak preserve regularizer, skipping Stage 1 forces it to learn latent specialization, injector contextualization, and downstream behavior jointly from random initialization, making optimization harder. This is supported by our ablations: collapsing the two memories into one shared bank drops accuracy from 53.84 to 47.53 (Fig. 3), and freezing the injector further reduces it to 50.44 (Table 2). These results indicate that DLMR benefits from both **memory specialization** and a **trainable interface**.
>
> We also remove Stage 1 and randomly initialize the latent banks before joint training with the injector in Stage 2. As shown in **Table R1**, This causes clear drops on both MMStar and MathVista, showing that Stage 1 is not merely a cosmetic warm-up, but a useful initialization for stable dual-memory specialization.
>
> |Modelvariant|MMStar|MathVista|
> |---|---:|---:|
> |withStage1|68.2|72.1|
> |randominit|65.1|69.4|
>
> We further analyze dataset-level router behavior using the visual-memory trigger ratio. As shown in **Table R2**, removing Stage 1 causes the router to trigger visual memory much less often on both MMStar and MathVista, suggesting that Stage 1 helps establish more stable visual-memory specialization and routing behavior.
>
> |Modelvariant|MMStar|MathVista|
> |---|---:|---:|
> |withStage1|0.5|0.4|
> |randominit|0.2|0.1|
>
> ### Response to W2/Q2
>
> We agree that token count is only a proxy for efficiency. Empirically, the added overhead of the Router + Injector + cache reset does not outweigh the benefit of reducing redundant decoding on the long-horizon workloads we evaluate. Appendix Table A1 shows that, on Qwen2.5-VL, DLMR improves accuracy from 68.3% to 71.4% on general tasks while slightly reducing latency from 5.6s to 5.4s, and on reasoning tasks also.
>
> A key reason is that the Router and Injector are lightweight LoRA-based modules, and DLMR injects memory only sparsely rather than at every decoding step. Specifically, injection is restricted to delimiter-eligible positions, capped by a per-sample maximum number of injections, and limited to a small discrete budget set. In addition, the router is trained with an efficiency-aware objective that prefers smaller budgets when the answer is correct. As a result, cache-reset overhead is bounded, while the gains arise from replacing redundant textual re-derivation with sparse latent reuse.
>
> We also report a decomposed efficiency analysis as follow.
>
> |Benchmark|length|injections|budget|inj(s)|cache(s)|decode(s)|total(s)|
> |---|---:|---:|---:|---:|---:|---:|---:|
> |MathVista|702|2.8|7.2|0.76|1.35|9.34|11.45|
>
> ### Response to Q3
>
> **Regarding the concern about input-agnostic global latents,** we do not use $Z^{(v)}$ and $Z^{(r)}$ as static instance memories, but as **shared memory bases** that are first contextualized by the current prefix through the injector, yielding **step-specific memory tokens** rather than directly reusing a fixed global cache. This design is further regularized by explicit visual/reasoning specialization in Stage 1 and selective, budgeted routing, which reduces the chance that the shared latents collapse into training-specific shortcuts. Empirically, we do not observe degraded generalization: DLMR improves consistently across multiple backbones in [Ana.1; table3] and also transfers well to **unseen benchmarks** in Appendix Table A2, outperforming baselines on LogicVista, OCR-Bench, and MMT-Bench. Therefore, we view the shared latents not as saturated global buffers, but as compact reusable priors whose actual contribution remains conditioned on the current input and decoding state.
>
> ### Response to W4/Q4
>
> To isolate extra compute from memory content, we added a matched baseline that injects **shared learnable static tokens** at the same routing positions and with the same token budgets, but without the context-dependent injector. On Qwen2.5-VL, DLMR still improves over this static-token baseline from **62.9 to 68.2 on MMStar** and from **69.1 to 72.1 on MathVista**, showing that the gain is not due to extra token depth alone. Instead, DLMR performs **state-conditioned memory reuse**: injected tokens are generated from the current decoding state via $g_\phi(E_t, Z^{(s)}_{1:k}, k)$, and the reasoning bank is explicitly trained to align with textual reasoning state rather than acting as fixed pause tokens.
>
> |Method|MMStar|MathVista|
> |---|---:|---:|
> |Sharedlearnablestatictokens|62.9|69.1|
> |DLMR|68.2|72.1|

---

> > ### Author Rebuttal · Reviewer_KSkE · 2026-04-02
> >
> > The rebuttal make sense to me. I would like to raise my score from 3 to 4.

---

> > > ### Author Response · Authors · 2026-04-02
> > >
> > > Thank you very much for the thoughtful follow-up and for reassessing the paper. We sincerely appreciate that you found our rebuttal helpful and that your concerns have been adequately addressed.

---

### Official Review · Reviewer_tVLa · 2026-03-10

**Soundness:** 4
**Presentation:** 3
**Significance:** 3
**Originality:** 3
**Overall Recommendation:** 4
**Confidence:** 5

**Summary:**

This paper studies long-horizon failures in multimmodal large language models, where visual evidenceand intermediate reasoning constraints fade during extended decoding. It proposes Dual-Latent Memory Routing (DLMR), a parameter-efficient module for frozen MLLMs that keeps two latent memory banks-one for visual information and one for reasoning state and uses a router to decide when and
how much memory to inject during generation. A lightweight injector turns the selected latent memories into context-aware tokens that can be reused on demand. The method is trained in three stages and evaluated on diverse multimodal reasoning benchmarks across multiple backbones, reporting improved accuraacy, better token efficiency, and transfer to text-only reasoning tasks.

**Compliance With Llm Reviewing Policy:**

Affirmed.

**Final Justification:**

The rebuttal strengthens the paper and addresses most of my concerns.

**Key Questions For Authors:**

1. What are the absolute trainable parameter counts for DLMR (including memories, injector LoRA, and router) as a fraction of each backbone?

2. How sensitive is performance to the delimiter set, the routing cap N_max, the memory sizes, and the choice of budget set K+?

3. Can you quantify the incremental decoding cost of mid-generation memory injection and KV-cache reset, ideally by isolating cache-reset overhead from the token savings achieved by DLMR?

4. Are there classes of tasks where DLMR underperforms the baseline or where memory injection harms performance?

**Limitations:**

yes

**Strengths And Weaknesses:**

S1. Clear motivation.
The paper identifies a credible long-horizon failure mode in MLLMs and motivates it with an intuitive attention-decay observation, clearly explaining why existing models struggle with extended reasoning.

S2. Practical design.
The method keeps the backbone frozen while introducing lightweight memory-related components, making it efficient to deploy and easy to adapt across different models.

S3. Strong empirical validation.
Experiments span multiple general and reasoning-heavy benchmarks, different backbones, and transfer settings. The largest gains appear on harder reasoning tasks, and ablation studies clarify the roles of disentangled memories, the trainable injector, and adaptive routing.

W1. Moderate novelty and heuristic design choices.
The main contribution—separating visual and reasoning memories—is interesting but incremental relative to prior memory-augmented MLLM work. Several components, such as the dual-memory split, delimiter-triggered routing, and always-on visual injection, appear heuristic and lack deeper theoretical or empirical justification.

W2.  Incomplete analysis of efficiency and limitations.
Claims about parameter efficiency are not well quantified, with limited reporting of trainable parameter counts or sensitivity to memory size and routing budgets. The runtime analysis also does not isolate the latency overhead introduced by KV-cache resets during mid-generation memory injection. Finally, the discussion of limitations, failure cases, and training costs remains relatively brief.

W3. Limitations analysis are thin.
The impact statement is present, but the paper offers limited discussion of failure modes, training cost, or when disentanglement might hurt.

---

> ### Author Rebuttal · Authors · 2026-03-31
>
> We thank the reviewer for the thoughtful and constructive feedback.
> ### Response to W1 / Q2
> We agree that DLMR builds on prior memory-augmented MLLMs, but its contribution is not merely adding memory. Our key novelty is an explicit memory interface that jointly models **what to reuse** (visual vs. reasoning memory), **how to inject it** (via a trainable injector), and **how much to allocate** (via state-dependent routing over discrete budgets), while keeping the backbone frozen. This design is motivated by the long-horizon failure mode in Sec. 3, where MLLMs progressively lose access to early visual evidence and intermediate constraints during decoding.
>
> **The dual-memory split is empirically necessary.** In Abl. 1 (Fig. 3), collapsing visual and reasoning memories into a single shared bank reduces average reasoning accuracy from 53.84 to 47.53. Freezing the injector further lowers accuracy to 50.44, indicating that the gains come not from extra memory tokens alone, but from disentangled memories together with a trainable interface.
>
> **Delimiter-triggered routing** is a practical mechanism for reducing routing frequency and controlling overhead. To test sensitivity, we compare no gating, the default \(D=\{, . \n\}\), and a coarser \(D=\{\n\}\) on MathVista and MMStar.As shown in **Table R1**, the default setting(ours) provides the best trade-off, achieving higher accuracy than no gating with shorter generations, while the coarser setting is slightly cheaper but less accurate.
>
> |Delimiter|MathVista|Avg.Len|MMStar|Avg.Len|
> |---|---:|---:|---:|---:|
> |No gating|69.9|942|64.7|661|
> |Coarser|70.2|722|67.1|524|
> |Default(ours)|72.1|746|68.2|533|
>
> **For the budget set $K^+$,** Abl. 3 (Fig. 4 and Tab. 4) shows that DLMR’s gains do not come from simply using larger budgets. With fixed budgets $k$, larger budgets increase generation length but do not improve accuracy: fixed \(k=8\) performs best among fixed-budget variants (52.71), while fixed \(k=16\) is longer yet slightly worse (52.04). Varying $k_{\max}$ further reveals different optima across task families: **general benchmarks peak at $k_{\max}=16$**, while **reasoning benchmarks peak at $k_{\max}=32$**. This indicates that DLMR learns **when** larger budgets are useful, rather than uniformly injecting more tokens.
>
> **For the routing cap $N_{\max}$,** we add a sensitivity study by varying the maximum number of routed injections per sample on MathVista. As shown in **Table R2**, a cap that is too small limits useful revisitation on long-horizon tasks, while larger caps yield diminishing returns at higher cost; the default setting provides the best balance.
>
> |$N_{\max}$|Avg.Acc|Avg.Len|
> |---|---:|---:|
> |N=3|71.6|647|
> |N=5|72.1|746|
> |N=8|71.8|847|
>
> **Prompt-end visual injection** is likewise a practical auxiliary design rather than the core novelty. It provides a stable perceptual anchor before autoregressive decoding, while routed injections handle later revisitation. Empirically, removing prompt-end injection causes a **modest but consistent drop**, reducing performance from **72.1 to 71.0** on **MathVista** and from **68.2 to 66.1** on **MMStar**, with the effect being more noticeable on perception-heavy/general benchmarks.
>
> ### Response to W2 / Q1 / Q3
>
> In DLMR, the backbone remains frozen throughout all stages; we only optimize lightweight memory-related modules: the dual latent banks, the injector-side projectors and LoRA adapters, and the router-side LoRA/head. Under our default setting in the paper and codebase (q_proj / v_proj, $M_v$, $M_r$, $\mathcal{K}^{+}$), this corresponds to **35.9M** trainable parameters in total, computed from the official backbone configs and our implementation. This is only **0.51\%** of Qwen2.5-VL-7B and **0.45\%** of InternVL3-8B. Stage 2 trains **30.83M** parameters (latents + injector), while Stage 3 trains only **5.07M** router parameters. We will add this absolute parameter breakdown explicitly in the revision.
>
> We also agree that the runtime analysis should better isolate the cost of mid-generation memory injection and KV-cache rebuild. The current paper already reports **competitive end-to-end wall-clock time**, and DLMR is often faster overall on long-horizon reasoning because it reduces redundant decoding (on Qwen2.5-VL-7B reasoning, **14.0s -> 11.5s**; on InternVL3-8B reasoning, **13.4s -> 13.1s**).
>
> ### Response to W3 / Q4
> On the evaluated benchmarks, we do not observe a systematic regime where DLMR underperforms the matched baseline; instead, its gains are largest on reasoning-heavy tasks where long-horizon forgetting is most pronounced. A likely **low-gain regime** is short-horizon or single-step tasks, where revisiting earlier evidence is less necessary. We also do not claim disentanglement is universally optimal: while it helps clearly on our benchmarks, for tasks with highly fused state, a stronger separation prior may reduce flexibility relative to a shared memory design.

---

> > ### Author Rebuttal · Reviewer_tVLa · 2026-04-02
> >
> > Thank you for the detailed and thorough rebuttal.
> >
> > The authors provide substantial additional evidence addressing my concerns. In particular, the ablations and sensitivity analyses clarify the role of the dual-memory design, routing mechanism, and budget choices, making the overall approach better justified empirically. The added parameter breakdown and efficiency discussion are also helpful and significantly strengthen the practical claims.
> >
> > While the method still involves several heuristic design choices and the novelty remains moderate, I find the empirical validation and clarifications sufficient to support the paper’s contributions.
> >
> > Overall, the rebuttal strengthens the paper and addresses most of my concerns.

---

> > > ### Author Response · Authors · 2026-04-02
> > >
> > > Thank you very much for the thoughtful follow-up and for taking the time to reassess the paper. We greatly appreciate that you found the additional ablations, sensitivity analyses, and efficiency discussion helpful in addressing most of your concerns and strengthening the paper.

---

### Official Review · Reviewer_oViH · 2026-03-12

**Soundness:** 3
**Presentation:** 3
**Significance:** 3
**Originality:** 3
**Overall Recommendation:** 5
**Confidence:** 3

**Summary:**

This paper aims to address the problem that MLLMs often suffer from "long-horizon forgetting" during complex tasks, as their monolithic growing context causes early visual cues and intermediate logic to fade over time. To address this, the authors propose a parameter-efficient Dual-Latent Memory Routing (DLMR) mechanism. Through a learned router and a lightweight injector, the system dynamically retrieves fading information from two disentangled latent pools (visual and reasoning). All these components serve as a small "probe" and are inserted selectively as a few compact "memory tokens" into the model's current decoding step. These injected tokens will have a very high attention score to previous visual tokens or reasoning tokens, like a "cheat sheet" to copy the fading long distant information to the current step to address the forgetting problem. The experimental results show that such tokens enhance both accuracy and token efficiency without altering the backbone model.

**Compliance With Llm Reviewing Policy:**

Affirmed.

**Key Questions For Authors:**

No questions.

**Limitations:**

The author discuss the limitations.

**Strengths And Weaknesses:**

**Strengths**
1. The motivation is strong and the proposed method is efficiency and effective to address the problem.
2. The experimental results show improved performance.

**Weaknessess**
1. Figure 2 is too complex and could be simplified.

---

> ### Author Rebuttal · Authors · 2026-03-31
>
> We thank the reviewer for the positive evaluation and encouraging comments. We are glad that the reviewer finds the motivation strong, the method effective/efficient, and the empirical results solid.
>
> We also agree that Figure 2 could be simplified. In the revision, we will improve its readability by reducing visual complexity and clarifying the training and inference workflow.

---

> > ### Author Rebuttal · Reviewer_oViH · 2026-04-04
> >
> > Thanks for the rebuttal, I decide to keep my positive score.

---

### Decision · Program_Chairs · 2026-04-30

**Decision:**

Accept (spotlight)

**Comment:**

After the author response, all four reviewers agree to accept the paper, and the AC concurs.  The reviewers appreciate the sound motivation and effective method, which achieves higher accuracy with fewer tokens.  The author response addressed most reviewer concerns, though some remain as articulated by 7Qnx.  The additional clarifications and ablations should be included in the final paper or supplemental.